# Convergent World Representations and Divergent Tasks

**Core Francisco Park** [1] [2]

## Abstract

While neural representations are central to modern deep learning, the conditions governing their geometry and their roles in downstream adaptability remain poorly understood. We develop a framework clearly separating the underlying world, the data generation process and the resulting model representations to study these questions in a controlled setup. 5,075 city coordinates define the world and 7 geometric tasks generate the training data for autoregressive training. We find that different tasks give rise to qualitatively and quantitatively distinct world representation geometries. However, multi-task training drives convergence of world representations: models trained on non-overlapping tasks develop aligned geometric representations, providing controlled evidence for the Multitask Scaling Hypothesis of the Platonic Representation Hypothesis. To study adaptation, we pretrain models on all tasks, then test whether *new* entities (cities) can be consistently integrated into the representation space via fine-tuning. Surprisingly, we find that despite multi-task pretraining, some tasks, which we call *divergent*, actively harm the representational integration of new entities and harm generalization. Our results show that training on multiple relational tasks reliably produces convergent world representations, but lurking divergent tasks can catastrophically harm new entity integration via fine-tuning.

## 1. Introduction

The nature of representations and mechanisms learned by deep neural networks, or in fact any intelligent system, and their relation to generalization is a central topic in deep learning research (Hubel & Wiesel, 1962; Rosenblatt, 1958; Fukushima, 1980; Rumelhart et al., 1986). Recent work has demonstrated that neural networks trained on vast amounts of data can capture diverse, disentangled and sometimes interpretable aspects of their training data, or even of the world underlying the data (Bengio et al., 2014). These rich representations are generally thought to underlie the generalization and adaptability of neural networks to unseen, out-of-distribution scenarios.

Recent work on internal representations of language models has provided evidence that neural networks can develop structured representations of the underlying data rather than merely memorizing surface patterns (Li et al., 2022; Gurnee & Tegmark, 2023; Nanda et al., 2023b).

However, major open questions remain. When interpretable representations are discovered in neural networks, it is often unclear whether their emergence is surprising or inevitable, what geometry they will take and how they support generalization. Even less understood is how these representations adjust during fine-tuning and downstream adaptation.

Answering these questions is difficult in real-world settings, where the key factors, the world, the data and the model, are entangled and costly to vary independently. In this work, we develop a synthetic framework where these factors can be precisely controlled and systematically studied.

**This work.** To study these questions, we decouple the underlying *world* from the *data generation process* to control them independently. Concretely, we adopt the coordinates of real-world cities as our "world," a ready-made complex structure with ground-truth geometry, and define 7 geometric tasks on top of it. We train autoregressive Transformers on this data and study how world representations form and vary across tasks, surfacing preliminary evidence for the Platonic Representation Hypothesis (PRH) (Huh et al., 2024). Crucially, this setup allows us to define consistent updates to the world (adding new cities) that produce predictable changes in the data, letting us test whether models can absorb such updates via fine-tuning. Our contributions are as follows:

- **A Framework Decoupling World, Data and Model. (Sec. 3)** We separate the underlying world (city coordinates) from the data generation process (7 geometric tasks), enabling systematic study of how different tasks

[1]Center for Brain Science, Harvard University, Cambridge, MA, USA [2]Prior Computers, Cambridge, MA, USA. Correspondence to: Core Francisco Park <corefranciscopark@g.harvard.edu>.

*Proceedings of the 43rd International Conference on Machine Learning*, Seoul, South Korea. PMLR 306, 2026. Copyright 2026 by the author(s).

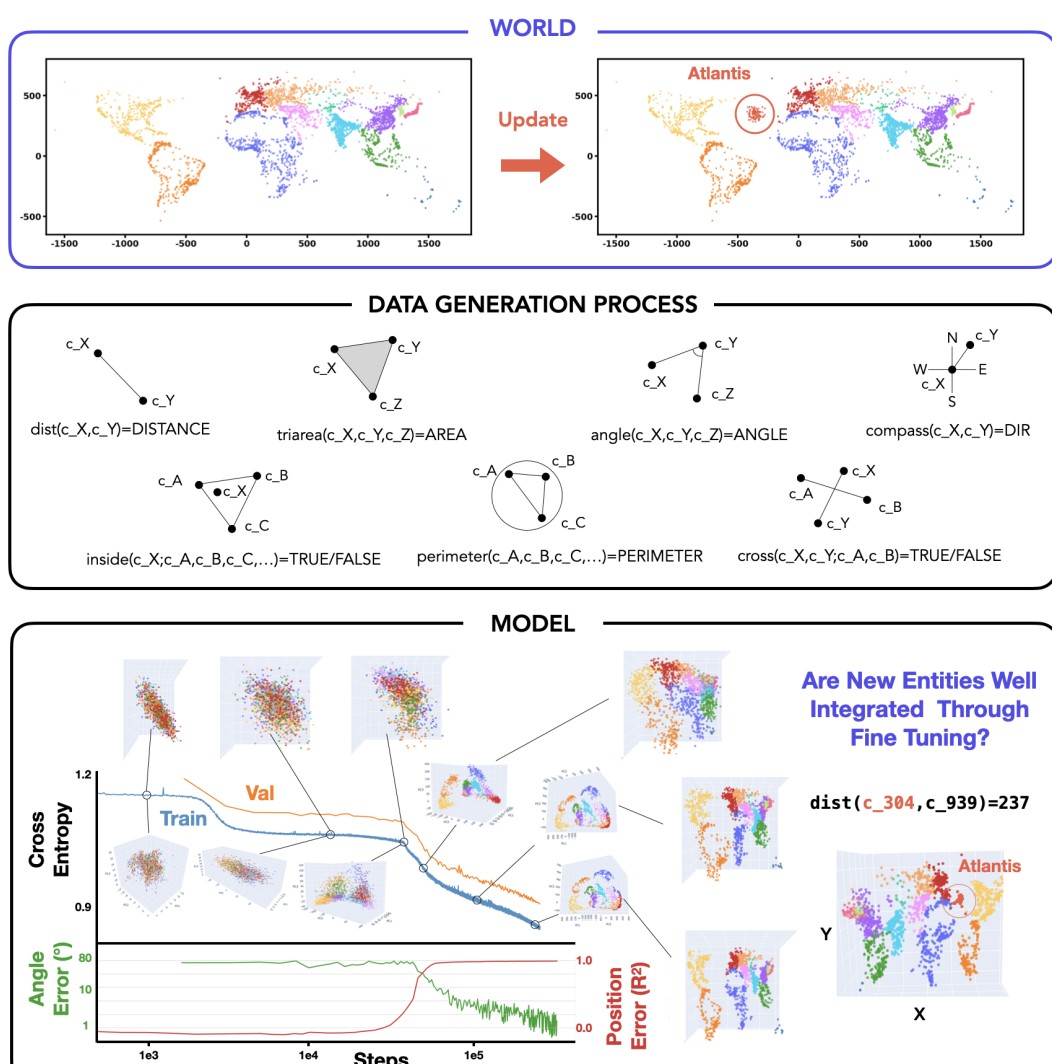

*Figure 1.* **Overview of the World-Data-Model framework. Top:** The world consists of 5,075 real city coordinates; we test adaptation by adding 100 synthetic `Atlantis` cities (App. B.1). **Middle:** Seven geometric tasks generate training data from city coordinates (App. B.2). **Bottom:** Training dynamics of one model, showing loss curves, linear probing accuracy for coordinate reconstruction and visualizations of internal representations (PCA and linear probe projections) at different training stages. See App. Fig. 9 for all training curves.

shape representations of the same world. The world provides ground-truth coordinates for directly assessing representation quality via probing. This setup also allows defining consistent world updates (adding synthetic `Atlantis` cities) to test whether models can adapt their representations accordingly.

- **Task-Dependent Geometry and Multi-Task Convergence. (Sec. 4)** We show that different tasks operating on the same world produce highly variable representational geometries across tasks and seeds. However, multi-task training drives convergence: models trained on multiple tasks show higher representational alignment, even when

they share no common tasks. This provides partial evidence for the Multitask Scaling Hypothesis, one proposed mechanism for the Platonic Representation Hypothesis.

- **Divergent Tasks Harm Fine-Tuning of New Entities Despite Multi-Task Pretraining. (Sec. 5)** We test whether models can integrate new entities (`Atlantis` cities) via fine-tuning. We find that single-task representational similarity (CKA) partially predicts cross-task generalization. In a multi-task fine-tuning setting, we find surprising "divergent" tasks which hinder integration of new entities into the learned manifold, actively harming generalization.

## 2. Related Work

**Internal Representations.** Recent work has revealed that language models develop structured world models encoding geographic, temporal and relational information (Li et al., 2022; Gurnee & Tegmark, 2023; Nanda et al., 2023b; Marks & Tegmark, 2024). Furthermore, PRH posits that diverse models converge toward similar representational structures (Huh et al., 2024), while recent work questions representational optimism in deep learning (Kumar et al., 2025). In this work, we study factors controlling representation formation and how networks integrate new entities via fine-tuning.

**Fine-tuning.** The pretraining-finetuning paradigm has become central to modern deep learning. Despite widespread success, fine-tuning exhibits poorly understood behaviors such as the reversal curse (Berglund et al., 2024) or emergent misalignment (Betley et al., 2025). Furthermore, careful studies of fine-tuning and other low-compute adaptation methods have raised pessimism about whether models can learn fundamentally new abilities, suggesting they may merely form "thin wrappers" around pretrained representations (Jain et al., 2023; Ward et al., 2025; Yue et al., 2025; Qin et al., 2025). Our work examines this question in a controlled setup where ground-truth world structure enables precise measurement of representation adaptation.

**Multi-task Learning.** Multi-task learning improves generalization through shared representations (Caruana, 1997); in some sense, modern foundation models represent an extreme form of multi-task training. Large-scale multi-task pretraining typically assumes rich representations emerge from data diversity (Aghajanyan et al., 2021), but the precise mechanisms remain underexplored. While recent work studies task diversity in controlled settings (Michaud et al., 2023; Zhang et al., 2025), the focus is on aggregate behaviors such as performance or scaling laws. Here, we define tasks as geometric functions over a shared world to investigate how task structure shapes *representations*.

**Synthetic Data.** The cost and complexity of foundation models has motivated synthetic approaches for controlled study of in-context learning (Xie et al., 2021; Chan et al., 2022; Reddy, 2023; Raventós et al., 2023; Park et al., 2024b; Wurgaft et al., 2025), compositional generalization (Okawa et al., 2024; Park et al., 2024c), grammar/knowledge acquisition (Allen-Zhu & Li, 2023b;a), and interpretability methods (Menon et al., 2025; Hindupur et al., 2025). Most relevant to our work, Jain et al. (2023) used synthetic data to argue fine-tuning creates only thin wrappers over pretrained capabilities, while Nishi et al. (2024) studied formation and destruction of representational structure. However, existing synthetic frameworks typically design data generation processes without explicitly distinguishing between the underlying world and how data is sampled from it. Our work introduces a framework that makes this distinction explicit, enabling systematic study of how different views of the same world shape neural representations and their downstream adaptability.

For further discussion, see App. A.

## 3. Experimental Framework: Decoupling World, Data and Model

Our framework uses geographic tasks where models solve geometric problems involving city coordinates. This naturally separates the underlying world (coordinates) from data generation (tasks), while providing ground-truth for measuring representation quality. Our framework provides three key properties:

1. **Learnability:** All tasks are deterministically generated from the same underlying coordinates. A model that learns the world structure can leverage it across all tasks.
2. **Latent State:** Models never see coordinates directly, only task outputs, allowing us to probe whether they internally reconstruct the world structure.
3. **Consistent Updates:** Modifying the world (e.g., adding new cities) produces self-consistent updates across all tasks, defining a *clear expectation* for what a model with proper world representations should internalize.

**Framework.** Let $\mathcal{W}$ denote a *world*: a set of entities $\{e_1, \ldots, e_N\}$ each with latent attributes $z_i \in \mathcal{Z}$. A *data generation process* is a set of tasks $\mathcal{T} = \{T_1, \ldots, T_K\}$, where each task $T_k : \mathcal{Z}^{n_k} \to \mathcal{Y}_k$ maps $n_k$ entity attributes to an output space $\mathcal{Y}_k$. Training data for task $T_k$ is generated by sampling entity tuples $(e_{i_1}, \ldots, e_{i_{n_k}})$ from $\mathcal{W}$ and computing $y = T_k(z_{i_1}, \ldots, z_{i_{n_k}})$.

A model $M$ observes only entity identifiers and task outputs, never the latent attributes $z_i$ directly. We say $M$ has learned a *world representation* if there exists a probe $P$ such that $P(M(e_i)) \approx z_i$ for all entities.

A *world update* $\mathcal{W} \to \mathcal{W}'$ (e.g., adding or modifying entities) induces consistent updates across all tasks by simply applying the same $T_k$ to the new or modified entities.

**Instantiation.** Concretely, our world consists of 5,075 real-world cities filtered by population $> 100{,}000$ (Fig. 1, top). We define 7 geometric tasks that take 2 or more city coordinates as input and compute a geometric value (Fig. 1, middle).

Each task query follows a structured format where city IDs (e.g., `c_1234`) serve as inputs to geometric functions, all character-tokenized for autoregressive prediction. For instance, `dist(c_0865,c_4879)=769` queries the distance between two cities, while

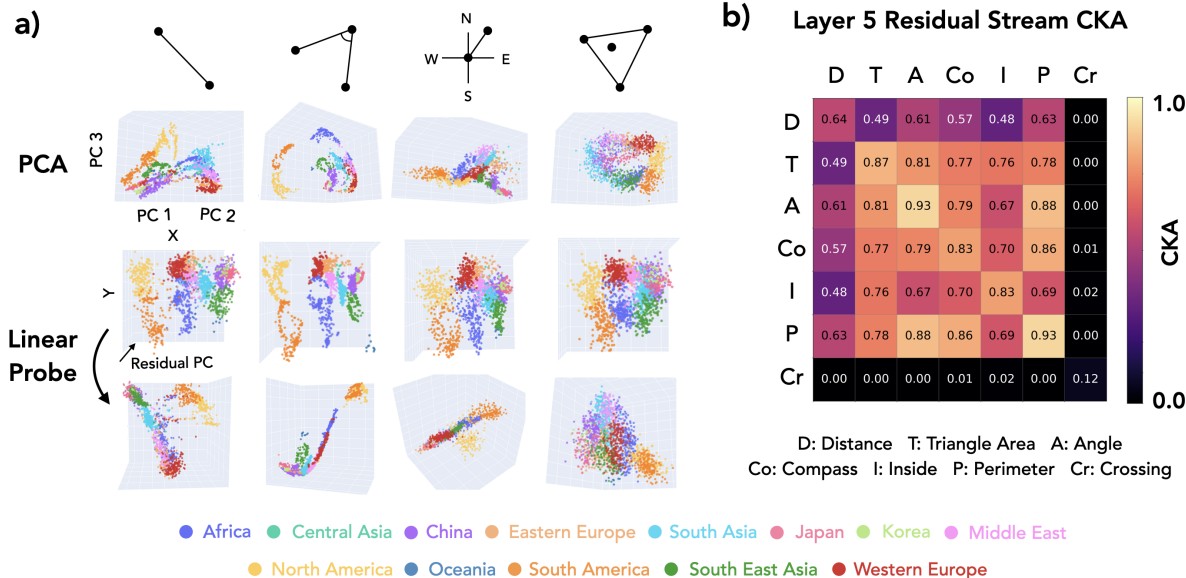

*Figure 2.* **World representation geometry depends on the data generation process.** (a) Different tasks create distinct geometries: `distance` (thread-like), `angle` (2D manifold), `compass` (fragmented), `inside` (diffuse). Row 1: PCA. Row 2: Linear probe projections. Row 3: Rotated views showing hidden structure. See App. Fig. 8 for more seeds. (b) CKA matrix at layer 5, estimated across 3 seeds. `Crossing` (Cr) fails to train alone. See App. Fig. 11 for SEM and layers 3, 4, 6. 3D visualizations: link (or Supp. Mat.).

`cross(c_2345,c_6789;c_0123,c_4567)=TRUE` checks whether two line segments intersect.

To test adaptation, we define `Atlantis`: 100 synthetic cities placed in the Atlantic Ocean. Models never observe `Atlantis` during pretraining; we use it in Sec. 5 to test whether fine-tuning can integrate new entities into world representations in a way that generalizes across tasks.

## 4. World Representations Converge Under Multi-Task Learning

We now study how task composition during pretraining shapes internal world representations by training Transformers on different task subsets and probing their representational geometry (see App. B.3 for training details).

**Result 1: World Representations Emerge through Autoregressive Training** We first demonstrate that world representations emerge through autoregressive training (Fig. 1, bottom). Training on the `angle` task, the model starts with random representations, goes through a loss plateau while clustering nearby cities, then forms world-aligned geometry as loss drops and task accuracy improves. The linear probe $R^2$ for coordinate decoding rises slightly before angle accuracy improves, reminiscent of hidden progress measures found during grokking (Nanda et al., 2023a). *Notably, once representational structure forms, it remains largely fixed for the remainder of training: representations are essentially fixed in the first ~15% of training, remaining static while loss continues to decrease and accuracy rises* (see App. 10

for visualization across tasks). This early saturation of representations echoes findings on critical learning periods in deep networks (Achille et al., 2019) and loss of plasticity in continual learning (Dohare et al., 2024). Overall, we find stable formation of internal world representations through pure autoregressive modeling. While the emergence of linearly decodable coordinates might be anticipated given the geometric nature of the task[1], it provides a useful validation of our framework and sets the stage for our main question: **How do different tasks shape these representations?**

**Result 2: Data Generation Process Controls World Representation Geometry** We train models from scratch for each of the seven tasks and visualize their representations in Fig. 2(a): PCA projections, linear probe reconstructions and rotated views.

Different tasks produce qualitatively distinct geometries: `distance` forms thread-like structures, `angle` forms 2D manifolds, `compass` forms fragmented clusters, and `inside` forms diffuse representations. These qualitative patterns are relatively consistent across random seeds (see App. D.1). Despite geometric differences, we can linearly decode (x,y) coordinates from most tasks (row 2), though some tasks (`angle`) yield cleaner reconstructions than others. The third row shows manually rotated views revealing that representations differ substantially in non-probe direc-

---

[1]We regard *linear* decodability of world representations as non-trivial (albeit expected). However, this is not the focus of our study.

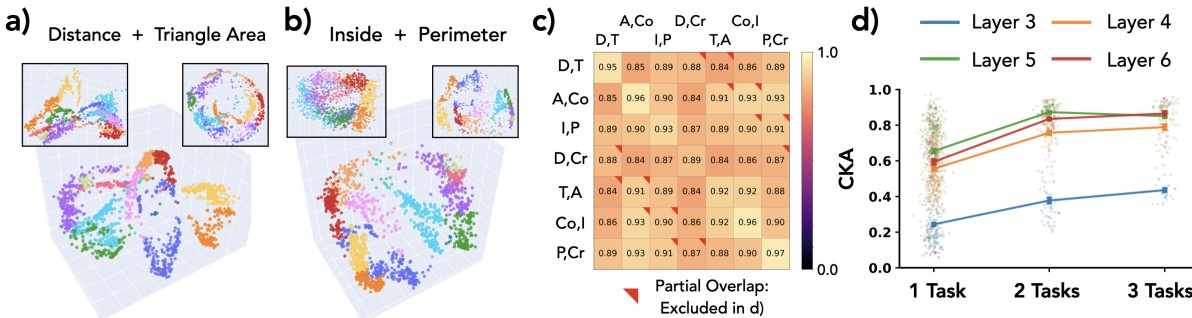

Figure 3. **Multi-task pretraining drives representational convergence.** (a,b) Two-task training creates more regular structures than single-task models. (c) CKA matrix (7×7) for two-task models shows higher alignment (see App. Fig. 12 for s.e.m.). (d) Average CKA increases with task count (1→2→3), saturating at ∼0.85 for layers 4-6 while layer 3 continues improving (see App. Fig. 13 for s.e.m.). `Crossing`, which failed to learn in single-task training, is excluded; including it would only strengthen the convergence finding.

tions, a reminder that *linear probing only surfaces what we look for.*

To establish a quantitative measure of representational similarity, we compute the CKA (Kornblith et al., 2019) (Fig. 2b). We find substantial variability even across seeds for the same task (see App. Fig. 11), but cross-task differences remain clear: `distance` produces particularly divergent representations, a result not obvious from PCA visualizations or from intuitions from the task. Note: the `crossing` task fails to train in isolation[2], explaining its near-zero CKA; intriguingly, it succeeds in multi-task settings (Result 3).

**Result 3: Multi-Task Learning Drives Representational Convergence** Having established that single-task training produces variable representations, we now ask: *does multi-task training reduce this variability?* This question connects to PRH (Huh et al., 2024), which observes that neural networks trained on diverse data develop aligned representations even across different modalities and architectures. One potential mechanism they suggest is the Multitask Scaling Hypothesis:

> *"There are fewer representations that are competent for N tasks than there are for M ≤ N tasks. As we train more general models that solve more tasks at once, we should expect fewer possible solutions."*

Our setup provides a controlled testbed for this hypothesis, with a ground-truth world model and multiple tasks defined over it. We trained models on selected two-task combinations (3 seeds each; see App. Fig. 14 for all 21 combinations). Fig. 3(a) shows representations when trained jointly

on `distance` and `triangle area` (with single-task versions shown for comparison), while (b) shows `inside` and `perimeter`. When trained on two tasks, models develop more regular representational structures.

We measure CKA between two-task trained models to quantify this alignment (Fig. 3(c)) and find it substantially higher than single-task models. One might expect high CKA when models share a task, but even models trained on *completely disjoint task pairs* show substantially higher alignment. In Fig. 3(d), we plot average CKA for models trained on 1, 2, and 3 tasks across layers 3-6, averaging only over models with completely disjoint task sets. Training on more tasks clearly leads to more aligned representations, with CKA saturating around 0.85 for 2 and 3 tasks in layers 4-6, while layer 3 continues improving. Notably, multi-task training also reduces per-seed variance of representations (App. Fig. 14b).

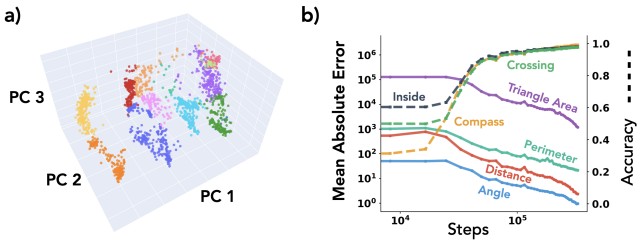

Figure 4. **7-task model.** (a) PCA projection of layer 5 representations naturally reveals world map structure. (b) Training curves showing successful learning of all 7 tasks, including `crossing` which failed in single-task training.

Overall, we find that **multi-task learning leads to convergent model internal representations**, providing controlled evidence for the Multitask Scaling Hypothesis explanation of PRH. Crucially, this alignment emerges even though single-task models achieve comparable task performance, all models reach high accuracy on their respective tasks. Since our networks are trained to representational conver-

---

[2]This likely connects to known hard-to-learn dynamics and gradient plateaus in training transformers (Pezeshki et al., 2021; Shah et al., 2020; Hoffmann et al., 2024; Bachmann & Nagarajan, 2025; Gopalani & Hu, 2025).

gence (as seen in Fig. 1), this demonstrates that the alignment is not simply a byproduct of optimization difficulty but rather that task diversity, not just data quantity or performance pressure, drives aligned representation learning.

An auxiliary finding: the `crossing` task, which was unlearnable alone, trains successfully when paired with any other task. We speculate that companion tasks provide structured coordinate representations that `crossing` can leverage, an implicit curriculum where easier tasks scaffold the learning of harder ones through shared representations.

To extend these findings, we trained a model on all 7 tasks simultaneously (Fig. 4). This model successfully learns all tasks, and its PCA projection naturally reveals the world map structure, approaching the perceived quality of linearly probed (x,y) coordinates without requiring any explicit coordinate supervision. Why multi-task training drives more linearly *surfaced* (i.e. higher magnitude) representations remains an open question worthy of future investigation. This 7-task model serves as the foundation for our fine-tuning experiments in the following section.

## 5. Divergent Tasks Harm Entity Integration via Fine-Tuning

In the previous section we observed how multi-task pretraining yields shared representations for different tasks. In this section, we investigate generalization properties of fine-tuning on top of such representations. However, unlike most fine-tuning studies which focus on changing model behavior in a certain way and evaluate generalization across entities, we study the inverse: fine-tuning an entity into the model and evaluate generalization across tasks. To this end, we use the 7-task model trained in the previous section (Fig. 4).

As mentioned in Sec. 3, we introduce 100 `Atlantis` cities to the world and fine-tune on data containing `Atlantis` to probe for generalization. We emphasize that the introduction of `Atlantis` cities keeps the original dataset fully consistent with the world. Moreover, task operations on `Atlantis` cities are well-defined in the same framework. If the model learned the true data generation process with properly factored representations, it should be able to integrate `Atlantis` seamlessly. If not, we suspect either the representations are fractured (Kumar et al., 2025) or gradient descent cannot trigger the right representational updates (Kumar et al., 2022).

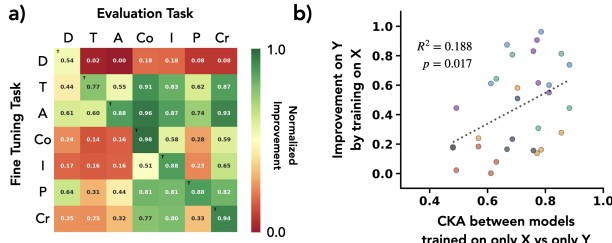

*Figure 5.* **Fine-tuning generalization and its correlation with representational similarity.** (a) Generalization matrix (averaged over 4 seeds; see App. Fig. 16 for individual seeds): each row is a model that integrated `Atlantis` via one task; columns show normalized improvement on `Atlantis` queries for each task (see App. C.1 for metric details). (b) For each task pair (X, Y), we plot the single-task CKA between X and Y against the normalized improvement on task Y after fine-tuning on task X (see App. Fig. 15 for an annotated version).

**Result 1: Pretraining Phase Representational Alignment Predicts Fine-Tuning Generalization *Despite* Joint Pretraining** We first address a simple question: when fine-tuning on `Atlantis` cities for a single task (e.g., `distance`), should we expect the model to automatically generalize to using `Atlantis` for all other tasks?

To answer this, we fine-tune on 100k examples of a single task that include `Atlantis` cities, mixed with original pretraining data to avoid catastrophic forgetting and a small multi-task elicitation set (see App. B.3 for details).

The resulting generalization matrix is shown in Fig. 5(a). This matrix reveals rich phenomenology: some tasks like `distance` show no cross-task generalization (`Atlantis` remains usable only for that task), while `angle` triggers significant generalization across all tasks. Intriguingly, we observe an apparent inverse relationship: tasks that efficiently trigger cross-task generalization of new entities are often those that don't easily benefit from other tasks' fine-tuning, though this relationship is noisy.

Unexpectedly, we find that *generalization performance correlates with the CKA values from single-task pretraining* (Result 2 of Sec. 4). This is puzzling: the CKA values come from models trained from scratch on individual tasks, yet they partially predict fine-tuning behavior of a model pretrained on all tasks jointly (Fig. 5b). *If the multi-task model truly uses unified representations for cities, why would single-task representational properties matter?*

For clarity, we define two terms: **Divergent tasks** are tasks which have low CKA compared to others when trained in isolation (in our case the `distance` task). **Hidden spaces** are representation spaces not surfaced by PCA or probing but used by divergent tasks.

We hypothesize:

*"Even though models develop joint world representations which converge in multi-task pretraining, gradient descent on divergent tasks might fail to act on these shared representations during fine-tuning, instead utilizing hidden spaces that don't propagate updates across tasks."*

Our question is then two-part:

1. To what extent do divergent tasks affect fine-tuning generalization?
2. Will gradient descent on divergent tasks fail to merge fine-tuning introduced concepts to the original representation manifold?

**Result 2: Divergent Tasks Catastrophically Harm Generalization**   To investigate how divergent tasks affect generalization, we move from single-task to multi-task fine-tuning settings. First, we introduce a simple heuristic model: fine-tuning on a concatenated dataset $\{D_1, D_2, ..., D_n\}$ (which do not provide conflicting supervision) should combine their individual effects. Specifically, when concatenating and shuffling all fine-tuning data to avoid sequential learning effects like catastrophic forgetting (McCloskey & Cohen, 1989), we suppose the improvement $\text{Imp}_i$ on task $i$ after training on tasks $j$ and $k$ to follow a **best-teacher model**:

$$\text{Imp}_i(D_j \cup D_k) = \max(\text{Imp}_i(D_j), \text{Imp}_i(D_k)) \quad (1)$$

To test this hypothesis, we fine-tuned the 7-task model on all $\binom{7}{2} = 21$ possible two-task combinations. Fig. 6(a,c) shows the *deviation* from our best-teacher expectation (averaged over 4 seeds; see App. Fig. 17 for raw improvements and App. Fig. 18 for individual seeds). Strikingly, we observe "red horizontal bands": **models that not only fail to benefit from multi-task training but actually perform worse than their best single-task component**. Notably, all these degraded performance bands involve the distance task. Fig. 6(c) quantifies this: when we split the deviation values into models with and without distance, we consistently observe lower-than-expected performance when the divergent task is included. This confirms that *divergent tasks (those with low single-task CKA) actively harm fine-tuning generalization rather than simply failing to contribute*. We next examine how this manifests in the learned representations.

**Result 3: Divergent Tasks Disrupt Representational Integration of New Entities**   Having shown that divergent tasks harm generalization (Question 1), we now address Question 2: does gradient descent on divergent tasks fail to merge new entities into the representation manifold?

We take two exemplars from the 21 fine-tuning runs: one without distance that generalized well (angle +

compass), and one with distance that was sabotaged (distance + perimeter). We first train a linear probe on the residual representations to predict x,y coordinates of all cities including Atlantis; these reconstructions are shown in Fig. 6(b) (top and bottom panels). In the well-integrated case, Atlantis cities lie within the world data manifold, but in the ill-integrated case, Atlantis cities are put *off* the manifold. While this difference appears subtle in 2D projections, the effect is dramatic in 3D, we strongly encourage readers to explore our interactive visualizations (or Supp. Mat.). Next, we train a linear probe on 4000 non-Atlantis cities and apply it to Atlantis representations (middle panels). In the well-integrated case, Atlantis cities (red-orange) are relatively well reconstructed compared to ground truth (black crosses); in the ill-integrated case, reconstruction fails completely.

We quantify this effect in Fig. 6(d), showing histograms of absolute coordinate reconstruction error. When Atlantis is integrated via fine-tuning partially on divergent task data (red), reconstruction errors are nearly an order of magnitude larger than when integrated via purely non-divergent tasks (blue). For reference, non-Atlantis cities (yellow, still held out from probe training) show low reconstruction error as expected. One might hypothesize that Atlantis's location in the middle of the ocean creates inherently difficult geometry. To test this, we pretrained a model with Atlantis included from the start (green line). In this case, Atlantis cities are reconstructed as well as any other city, confirming that the integration failure stems from divergent task fine-tuning dynamics rather than geographic peculiarity.

*This suggests that divergent tasks cause optimization to encode new entities in hidden spaces rather than integrating them into the existing world manifold, explaining their failure to support cross-task generalization.*

We clarify that our findings are correlational: we do not yet know if interventions increasing single-task CKA would necessarily improve fine-tuning generalization. Rather, we identify representational divergence as a diagnostic marker for tasks that will harm multi-task fine-tuning performance.

Putting these results together: single-task representational divergence weakly predicts fine-tuning generalization even after joint pretraining, and the most divergent task (distance) actively harms integration of new entities. This raises a hypothesis: **certain task-architecture pairings may have intrinsic properties that induce gradient dynamics bypassing shared representations, causing updates in hidden subspaces that harm generalization, even when the network seem to have unified representations for the forward pass.**

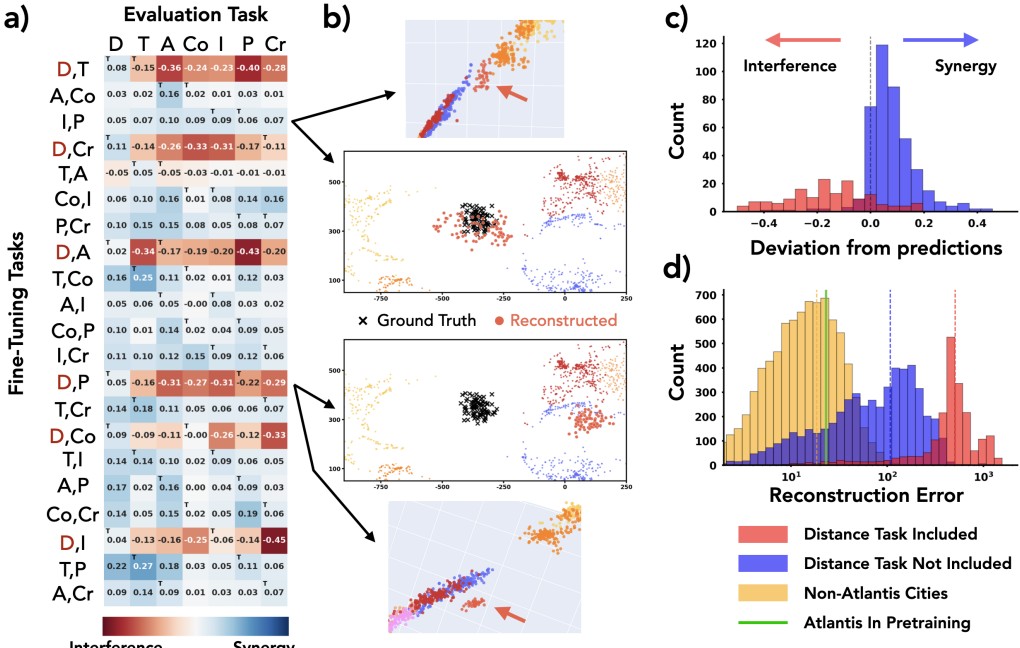

*Figure 6.* **Divergent tasks harm multi-task fine-tuning and disrupt representational integration.** (a) Deviation from best-teacher expectation for 21 two-task models (rows) across 7 evaluation tasks (columns), computed in normalized improvement space (see App. C.1); "red horizontal bands" show `distance` task combinations degrade performance below single-task baselines. (b) Representation visualization and linear probe reconstruction of `Atlantis`. (c) Histogram of deviation values: models including `distance` vs. not. (d) Linear probe `Atlantis` coordinate reconstruction error for models with `distance`, without `distance`, and baseline on pretraining cities; green vertical line indicates performance when `Atlantis` is part of pretraining. 3D visualizations: link (or Supp. Mat.).

## 6. Discussion

**Forward and backward modularity.** Our results highlight a distinction that is often overlooked: *modularity in the forward pass does not imply modularity in the backward pass.* Multi-task training produces clean, structured representations that can be easily decoded into world coordinates, yet these world models can be fractured and partial when it comes to adaptation. Gradient descent may not respect the forward-pass modularity when updating weights: fine-tuning on divergent tasks routes updates through pathways that bypass the shared world manifold, encoding new entities in task-specific subspaces.

**Continual learning and world models.** For truly general intelligence, internal world models should not only represent current state but adapt consistently when the world changes. Such adaptation is non-trivial: a single change can require cascading updates across tasks. Recent language models sidestep persistent adaptation via in-context learning, forming task-specific representations on the fly (Brown et al., 2020; Park et al., 2024a; Li et al., 2025b). However, fine-tuning consistently underperforms ICL for knowledge integration (Lampinen et al., 2025; Park et al., 2025). Our study grounds these questions in a controlled setting where we can measure whether gradient descent achieves consistent integration of new entities into existing representations.

**Dynamics of representations.** Most recent work on neural representations examines pretrained networks or their formation during a single pretraining run. There is growing interest in how representations change during adaptation, both at inference (Park et al., 2024a; Li et al., 2025b; Shai et al., 2025; Lubana et al., 2025; Bigelow et al., 2025) and during fine-tuning (Wang et al., 2025; Minder et al., 2025; Casademunt et al., 2025). To study representational adaptation rigorously, one must define both an updatable world and how updates to it propagate into training data. Our framework provides exactly this: introducing `Atlantis` defines how representations should update across all tasks.

**Future work.** Understanding the mechanistic basis of task divergence is an important open question. If divergence is a property of task-architecture pairing rather than learned weights, it may be predictable from task structure and gradient geometry alone, enabling identification of harmful tasks before training.

**Limitations.** We study representation formation in a controlled synthetic setting with small-scale models; generalization to large-scale natural settings remains unclear. We identify divergence as a diagnostic marker but do not reveal underlying mechanisms. Our PRH claims are partial, as we study only a single architecture and modality.

## Impact Statement

This paper presents work whose goal is to advance the field of Machine Learning. There are many potential societal consequences of our work, none which we feel must be specifically highlighted here.

## Acknowledgements

CFP thanks Andrew Lee, Thomas Fel, Ishaan Chandok, Kento Nishi, Hidenori Tanaka, Sunny Qin, Zechen Zhang, Felix Sosa, Helena Casademunt, Hugh Van Deventer, Itamar Pres, Fuming Yang, Sophie Wang, Andrew Lampinen, Vaishnavh Nagarajan, Venkatesh Murthy and Aravinthan Samuel for useful discussions and research support.

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

# Appendix

## A. Extended Related Work

**Internal Representations.** Understanding internal representations has roots in neuroscience (Hubel & Wiesel, 1962), informing early neural network development (Fukushima, 1980; Bengio et al., 2014; Rosenblatt, 1958; Rumelhart et al., 1986). Recent work has revealed that language models develop structured "world models" encoding geographic, temporal and relational information (Li et al., 2022; Gurnee & Tegmark, 2023; Nanda et al., 2023b; Marks & Tegmark, 2024), with similar representations emerging during in-context learning (Vafa et al., 2025). Mechanistic interpretability and sparse autoencoders have enabled decomposition of neural activations into interpretable features (Anthropic AI, 2023; Templeton et al., 2024). Researchers have also uncovered that models represent meaningful properties of data—concepts (Pearce et al., 2025; Higgins et al., 2017), features (Olah et al., 2017), and abstractions (Lee et al., 2025; Arditi et al., 2024)—in interpretable ways. Furthermore, PRH posits that diverse models converge toward similar representational structures (Huh et al., 2024). However, recent work questions this representational optimism, suggesting that deep network representations may be more brittle than previously assumed (Kumar et al., 2025). Only recent work has begun examining how representations emerge during pretraining in real LLMs (Li et al., 2025a; Ge et al., 2025) or how they change during fine-tuning (Lee et al., 2024). Our work takes a complementary perspective, studying the factors that control the formation of these representations and how networks integrate new entities into their representation space via fine-tuning.

**Fine-tuning.** The pretraining-finetuning paradigm has become central to modern deep learning, with seminal works establishing its effectiveness in computer vision (Krizhevsky et al., 2012; He et al., 2015) and natural language processing (Devlin et al., 2018; Radford et al., 2018). Despite widespread success, fine-tuning exhibits poorly understood behaviors such as the reversal curse (Berglund et al., 2024; Lampinen et al., 2025), out-of-context reasoning limitations (Treutlein et al., 2024), and off-target effects (Betley et al., 2025). On this background, careful studies of fine-tuning and other low-compute adaptation methods have raised pessimism about whether models can learn fundamentally new abilities, suggesting they may merely form "thin wrappers" around pretrained representations (Jain et al., 2023; Ward et al., 2025; Yue et al., 2025; Qin et al., 2025; Zhao et al., 2025; Zweiger et al., 2025). Fine-tuning has also been studied across diverse directions: parameter efficiency (Hu et al., 2021; Lester et al., 2021), zeroth-order optimization (Malladi et al., 2024), weight composition (Ilharco et al., 2023), and representation adaptation (Wu et al., 2024). Work on feature distortion (Kumar et al., 2022) is perhaps most related to ours, though representational changes are assumed rather than directly measured. Our work examines this question in a controlled setup where ground-truth world structure enables precise measurement of representation adaptation.

**Multi-task Learning.** Multi-task learning has long been studied as a way to improve generalization through shared representations (Caruana, 1997); in some sense, modern foundation models represent an extreme form of multi-task training. Large-scale multi-task pretraining typically assumes rich representations emerge from data diversity (Aghajanyan et al., 2021), but the precise mechanisms remain underexplored. Recent work has begun studying task diversity in controlled settings (Michaud et al., 2023; Zhang et al., 2025), though most studies still focus on aggregate behaviors such as capabilities and scaling laws rather than characterizing tasks or the knowledge they operate on. Our framework explicitly defines tasks as geometric functions over a shared world, enabling direct investigation of how task structure shapes representations.

**Synthetic Data.** The cost and complexity of foundation models has motivated synthetic approaches for controlled study of in-context learning (Xie et al., 2021; Chan et al., 2022; Reddy, 2023; Raventós et al., 2023; Park et al., 2024b; Wurgaft et al., 2025), compositional generalization (Okawa et al., 2024; Park et al., 2024c), grammar/knowledge acquisition (Allen-Zhu & Li, 2023b;a), and interpretability methods (Menon et al., 2025; Hindupur et al., 2025). Most relevant to our work, Jain et al. (2023) used synthetic data to argue fine-tuning creates only thin wrappers over pretrained capabilities, while Nishi et al. (2024) studied formation and destruction of representational structure. However, existing synthetic frameworks typically design data generation processes without explicitly distinguishing between the underlying world and how data is sampled from it. Our work introduces a framework that makes this distinction explicit, enabling systematic study of how different views of the same world shape neural representations and their downstream adaptability.

**Dynamics of Representations.** Recent work has begun studying how representations evolve during in-context learning (Shai et al., 2025; Demircan et al., 2024) or fine-tuning (Casademunt et al., 2025; Minder et al., 2025). Relatedly, Lubana et al. (2025) show that representations exhibit rich temporal dynamics that standard interpretability methods (e.g., SAEs) fail to capture due to stationarity assumptions. Fu et al. (2025) show that VLMs trained by merging LLMs and vision encoders often fail to utilize representations surfaced by the vision encoder, i.e. the representations exist but remain unused.

**Geometric Deep Learning.** Geometric deep learning studies how data geometry interacts with model architectures,

developing equivariant networks that respect symmetries (Bronstein et al., 2021; Cohen & Welling, 2016; Weiler & Cesa, 2021). While our world is defined on a 2D plane, one might ask: why not a sphere, torus, or other manifold? This is an interesting direction, but not our focus. We study how neural networks adapt internal representations to tasks in an arbitrarily chosen geometry. Moreover, a change in world geometry can be absorbed into the task definition (e.g., geodesic vs. Euclidean distance), so the key question remains how representations form given the task, not the underlying manifold. Planar coordinates also allow clean linear probing of world representations. Our models are standard transformers without geometric priors; we study what representations emerge purely from training on task data, treating geometry as emergent rather than imposed.

**Loss Plateaus.** Our `crossing` task fails to learn in single-task training despite escaping an initial plateau (likely output format learning), suggesting it remains stuck in a deeper plateau. Such plateaus are notoriously difficult for transformers. Recent work has studied this phenomenon mechanistically in transformers (Hoffmann et al., 2024; Gopalani & Hu, 2025; Singh et al., 2024), while others relate it to more general optimization challenges in deep learning such as simplicity bias and gradient starvation (Shah et al., 2020; Pezeshki et al., 2021; Bachmann & Nagarajan, 2025). Most related to our findings, Kim et al. (2025) show that multi-task training shortens loss plateaus, similar to why our `crossing` task trains successfully when joined with any other task.

## B. Experimental Details

This section provides detailed information about the world, data generation process, model architecture and training procedures used in our experiments.

### B.1. World

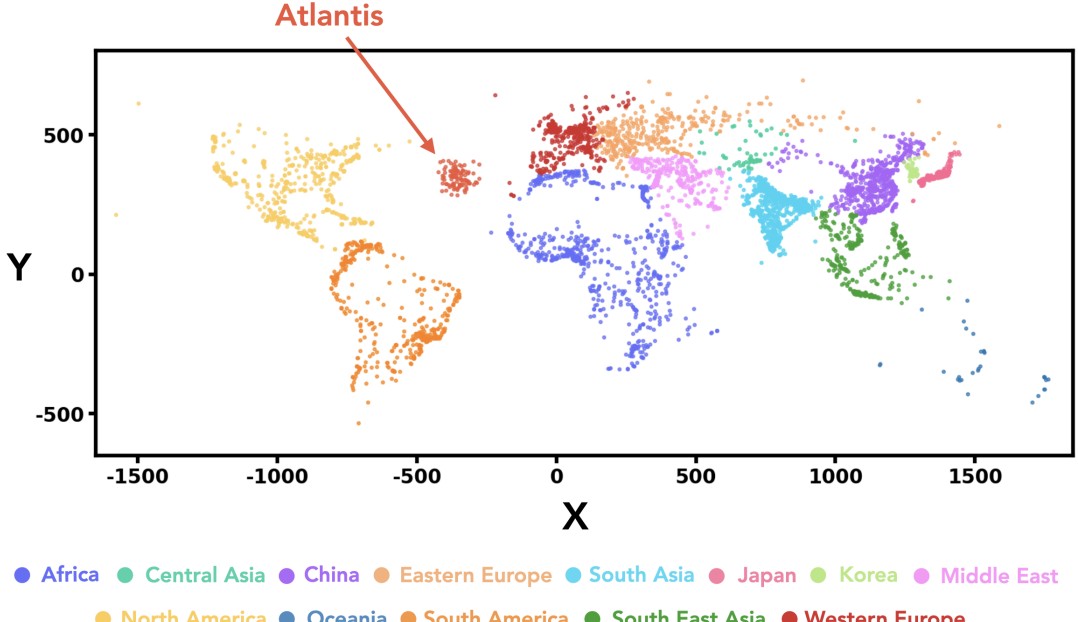

*Figure 7.* **Geographic distribution of cities used in our experiments.** 5,075 real-world cities plus 100 synthetic `Atlantis` cities (5,175 total). Cities span all continents and provide a fixed, measurable world structure. Coordinates use an equirectangular projection: $x = 10 \times$ longitude, $y = 10 \times$ latitude (in degrees). The `Atlantis` region (Atlantic Ocean) is used for out-of-distribution testing.

Our experiments use a geographic world consisting of 5,075 cities extracted from the GeoNames database (OpenDataSoft / GeoNames, 2025) with population greater than 100,000. Cities are distributed across all continents. This choice provides natural variation in density (e.g., dense regions like India versus sparse Oceania) that creates interesting computational challenges.

While we use real city coordinates, this work studies abstract geometric reasoning rather than actual geography, we project coordinates to Euclidean space using an equirectangular projection (as described above) and treat all tasks as pure geometry

problems.

Additionally, we introduce 100 synthetic `Atlantis` cities positioned in the Atlantic Ocean, centered at (longitude $-35°$, latitude $35°$) and following a Gaussian distribution with standard deviation of $3°$. These synthetic cities enable controlled out-of-distribution experiments, as models never observe `Atlantis` during pretraining but must generalize to it during evaluation. City IDs are randomly assigned from the range $[0, 9999]$, creating a sparse identifier space that models must learn to map to coordinates.

### B.2. Data Generation Process

**Tasks** We implement 7 geometric tasks that operate on city coordinates. All tasks use a consistent format: `task(arguments)=answer`, where city IDs are prefixed with `c_`. Numerical outputs (distance, area, angle, perimeter) are rounded to integers. Table 1 summarizes the tasks.

| Task | Input | Output Type | Unit/Values | Example |
|------|-------|-------------|-------------|---------|
| `distance` | 2 cities | Numerical | Scaled coords | `dist(c_865,c_4879)=769` |
| `triarea` | 3 cities | Numerical | Scaled coords$^2$ | `triarea(c_1234,c_5678,c_9012)=45823` |
| `angle` | 3 cities | Numerical | Degrees (0–180) | `angle(c_2345,c_6789,c_123)=97` |
| `compass` | 2 cities | Categorical | 8 directions | `compass(c_1234,c_5678)=NE` |
| `inside` | $1 + n$ cities | Categorical | TRUE/FALSE | `inside(c_9012;c_3456,...)=FALSE` |
| `perimeter` | $n$ cities | Numerical | Scaled coords | `perimeter(c_4567,c_8901,...)=2856` |
| `crossing` | 4 cities | Categorical | TRUE/FALSE | `cross(c_2345,c_6789;c_123,c_4567)=TRUE` |

*Table 1.* Summary of 7 geometric tasks. Numerical outputs are integers; "scaled coords" refers to the $\times 10$ coordinate system (Sec. B.1). Categorical tasks have discrete outputs: `compass` uses 8 cardinal directions (N, NE, E, SE, S, SW, W, NW), while `inside` and `crossing` are binary. The `inside` task tests if the first city lies within the convex hull of the remaining cities; `crossing` tests if line segment $(c_1, c_2)$ intersects segment $(c_3, c_4)$.

**Dataset Sizes** Each pretraining set consists of 1M rows of data per task. For fine-tuning, the dataset consists of: (1) 100k rows of the target task containing at least one `Atlantis` city, (2) 20k rows randomly sampled from the original pretraining data to prevent catastrophic forgetting, and (3) 256 rows per task (without `Atlantis`) to elicit multi-task performance.

### B.3. Model and Training

**Tokenization** We use character-level tokenization with 98 ASCII tokens (excluding space, which serves as the delimiter), plus special tokens for beginning-of-sequence (BOS), end-of-sequence (EOS) and padding (PAD).

**Architecture** We use the Qwen2(Yang et al., 2024) decoder-only transformer architecture with hidden size 128, 4 attention heads and 6 layers.

**Pretraining** We train models autoregressively on the full sequence (no prompt masking). All pretraining runs see 42M rows regardless of dataset size (e.g., 42 epochs for 1M rows, 6 epochs for 7M rows). Table 2 summarizes the hyperparameters.

**Fine-Tuning** Fine-tuning starts from the final pretrained checkpoint. We use a reduced learning rate of $1 \times 10^{-5}$ ($30\times$ smaller than pretraining) to avoid catastrophic forgetting. The fine-tuning dataset consists of 100k rows per task containing at least one `Atlantis` city. We train for 30 epochs with batch size 128.

## C. Analysis Methods

### C.1. Evaluation

**Generation Protocol** For evaluation, we use teacher forcing up to the "=" sign (the prompt), then generate autoregressively at temperature zero until reaching the EOS token or a maximum of 128 tokens (sufficient for all tasks).

| Hyperparameter | Value |
|---|---|
| Optimizer | AdamW |
| Learning rate | $3 \times 10^{-4}$ |
| Weight decay | 0.01 |
| Scheduler | Linear with warmup |
| Warmup steps | 50 |
| Batch size | 128 |
| Max sequence length | 256 |
| Total training rows | 42M |
| Initialization scale | 0.1 (std) |

*Table 2.* **Pretraining hyperparameters.**

**Task-Specific Metrics**   Categorical tasks (`compass`, `inside`, `crossing`) are evaluated using accuracy. Numerical tasks are evaluated using absolute error.

**Normalized Improvement**   To compare generalization across tasks with different metrics and scales, we define a normalized improvement score that maps performance to $[0, 1]$, where 0 indicates no improvement over the `Atlantis` baseline (before fine-tuning) and 1 indicates matching the pretrained model's performance on standard cities.

For **error-based tasks** (`distance`, `triarea`, `angle`, `perimeter`), where lower is better:

$$\text{NI} = \frac{\log(\text{baseline}_{\text{atlantis}}/\text{error})}{\log(\text{baseline}_{\text{atlantis}}/\text{baseline}_{\text{standard}})} \tag{2}$$

The logarithmic scaling ensures multiplicative improvements are treated equally (e.g., reducing error from 1000 to 100 is weighted the same as 100 to 10).

For **accuracy-based tasks** (`compass`, `inside`, `crossing`), where higher is better:

$$\text{NI} = \frac{\text{accuracy} - \text{baseline}_{\text{atlantis}}}{\text{baseline}_{\text{standard}} - \text{baseline}_{\text{atlantis}}} \tag{3}$$

Note that normalized improvement can slightly exceed 1.0 if, by chance, `Atlantis` cities perform better than the average pretrained city on some task.

### C.2. Representation Extraction

We extract representations from the residual stream after transformer blocks, specifically at layers 3, 4, 5, and 6 of our 6-layer model. Unless otherwise specified, all representation analyses in this paper use layer 5 representations.

To extract city representations, we pass a task prefix followed by a city ID through the model. For single-task models, we use the corresponding task prefix. For multi-task models (2-task and 3-task), we use the first task in the combination as the prefix. We verified that the choice of task prefix has negligible effect on the extracted city representations.

For a city with ID 1234, the input sequence is:

```
<bos> d i s t ( c _ 1 2 3 4 ,
```

We extract and concatenate the representations of two tokens: (1) the last digit of the city ID and (2) the following delimiter token (typically a comma). This yields a 256-dimensional representation ($128 \times 2$) per city, which we use for both PCA visualization and linear probing.

**Omitting cities with leading zeros**   We omit cities with IDs starting with 0, 00, or 000 from representation analyses. These cities form distinct clusters in representation space, separate from cities with IDs starting with non-zero digits. We

hypothesize this occurs because the digit 0 has special semantic status: in numerical outputs (distances, angles, areas), leading zeros never appear (e.g., "=769" not "=0769"), so the model learns to treat 0 differently when it appears as a leading digit. When 0 appears at the start of a city ID, the model may encode a feature indicating "this is an identifier, not a number," causing these cities to cluster separately. To ensure consistent evaluation across all cities, we exclude IDs matching the pattern `^[0][0-9]*$` (i.e., any ID starting with zero).

### C.3. Linear Probing & PCA

We use the representations described in Sec. C.2 for both PCA visualization and linear probing.

**Linear Probing**    We train linear probes to predict city coordinates $(x, y)$ from the 256-dimensional representations. We use a train/test split of 3250/1250 cities, training separate probes for $x$ and $y$ coordinates via ordinary least squares (OLS) without regularization. We report $R^2$ scores and mean absolute error in scaled coordinate units.

**PCA**    For visualization, we apply PCA to the representations and plot the first two or three principal components. We use consistent color coding based on geographic region to enable visual comparison across models and seeds.

**Reconstruction Error**    To quantify how well new entities (`Atlantis` cities) are integrated into the learned manifold, we train linear probes exclusively on non-`Atlantis` cities and evaluate reconstruction error on held-out `Atlantis` representations. Reconstruction error is measured as the absolute Euclidean distance between predicted and true coordinates. Large reconstruction errors indicate that new entities are encoded in different subspaces than the original cities.

### C.4. Centered Kernel Alignment

We use Centered Kernel Alignment (CKA) (Kornblith et al., 2019) to measure representational similarity between models. Given two representation matrices $X \in \mathbb{R}^{n \times d_1}$ and $Y \in \mathbb{R}^{n \times d_2}$ (same $n$ cities, potentially different dimensions), we compute linear kernel matrices $K = XX^T$ and $L = YY^T$, center them, and compute:

$$\mathrm{CKA}(X, Y) = \frac{\langle K, L \rangle_F}{\|K\|_F \|L\|_F} \tag{4}$$

where $\langle \cdot, \cdot \rangle_F$ denotes the Frobenius inner product. CKA yields a similarity score in $[0, 1]$ that is invariant to orthogonal transformations and isotropic scaling.

For each pair of models, we extract city representations (Sec. C.2) and compute CKA between the resulting matrices. We filter cities to exclude `Atlantis` and IDs starting with zeros. We report CKA values at layers 3, 4, 5, and 6, with layer 5 as the default unless otherwise specified.

## D. Additional Results

### D.1. Qualitative Representations

Fig. 8 shows PCA projections of city representations for single-task models across three random seeds (rows). The `distance` task consistently produces characteristic thread-like structures. `Angle` and `perimeter` often form larger 2D manifold-like structures. `triangle area` tends to produce arc-shaped geometries. `Compass` forms local clusters corresponding to directional categories, while `inside` produces a more global, diffuse structure.

While there is some seed-to-seed variability within each task, the broader categories remain distinguishable: `distance` representations are qualitatively distinct from the cluster-based representations of `compass` and `inside`, and both differ from the manifold-like structures produced by `triangle area`, `angle`, and `perimeter`.

### D.2. Training Dynamics

Fig. 9 shows training dynamics for all seven single-task models. Each panel displays three rows of metrics over gradient steps: (top) training and validation loss, (middle) task-specific performance metric alongside linear probe $R^2$ for coordinate decoding, and (bottom) linear probing distance error measuring how accurately city coordinates can be reconstructed from representations.

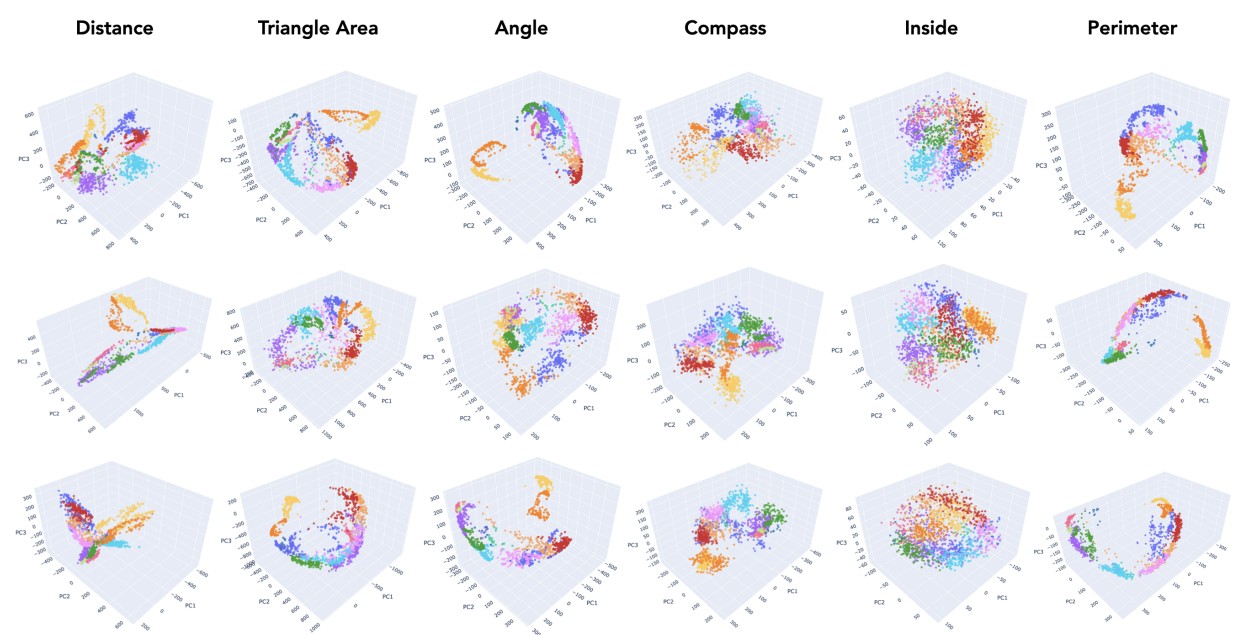

*Figure 8.* **Representation visualizations for single-task models across multiple seeds.** Each column shows a different task; each row shows a different random seed. Cities are colored by geographic region.

Several patterns emerge across tasks. First, all tasks except crossing eventually achieve high coordinate $R^2$ (red curves reaching $\sim$1.0), indicating that world representations form reliably across diverse geometric objectives. Second, the relationship between loss, task performance, and coordinate decodability varies across tasks. Third, crossing (panel g) fails entirely in single-task training. Loss remains high, accuracy stays near chance, and coordinate $R^2$ never rises, consistent with the main text observation that this task requires multi-task scaffolding.

**Representation Dynamics.** Fig. 10 visualizes how internal representations evolve during training via PCA projections at six checkpoints. A striking pattern emerges: once a representational structure forms, it remains largely fixed throughout the subsequent training phase where task accuracy continues to improve. Examining the gradient steps, representations are essentially fixed in the first $\sim$15% of training, remaining static while loss slowly decreases and accuracy rises. The distance task (top row) establishes its thread-like structure early; angle (middle row) settles into a 2D manifold; compass (bottom row) forms fragmented regional clusters, all within the first few checkpoints, with minimal subsequent change. What determines when representations stop evolving remains unclear, though it appears correlated with the initial loss drop. This may relate to recently observed gradient dynamics in language model training, where loss deceleration phases exhibit qualitatively different learning behavior (Mircea et al., 2025).

### D.3. Additional CKA Results

**Single-Task CKA Across Layers.** Fig. 11 shows CKA matrices for single-task models at layers 3, 4, 5, and 6. Each cell shows mean $\pm$ SEM across 3 seeds. We observe: (1) CKA values increase from layer 3 to layers 4–6, indicating that world representations become more consistent in later layers; (2) the distance task (D) shows lower CKA with other tasks across all layers, consistent with its divergent representational geometry; (3) crossing (Cr) shows near-zero CKA due to training failure in single-task settings; (4) diagonal entries (same task) can show significant variability, indicating that even identical training objectives can yield different representational solutions.

**Two-Task CKA.** Fig. 12 shows the CKA matrix for two-task models at layer 5. Compared to single-task models (Fig. 11, layer 5), two-task training substantially increases representational alignment: all off-diagonal entries exceed 0.84, compared to values as low as 0.48 for single-task models. Notably, diagonal entries (same task combination, different seeds) show

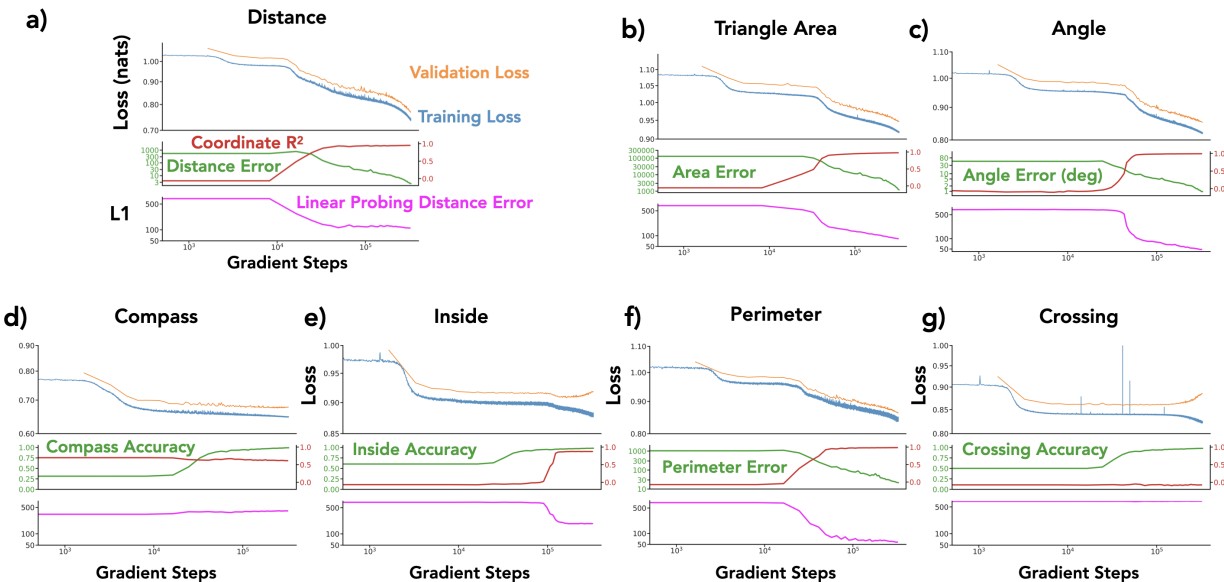

*Figure 9.* **Training dynamics for all single-task models.** (a) `distance`, (b) `trianglearea`, (c) `angle`, (d) `compass`, (e) `inside`, (f) `perimeter`, (g) `crossing`. Each panel shows three rows: (top) training loss (blue) and validation loss (orange); (middle) task-specific metric (green, left axis) and linear probe coordinate $R^2$ (red, right axis); (bottom) linear probing distance error (magenta). All plots use log-scale x-axis for gradient steps.

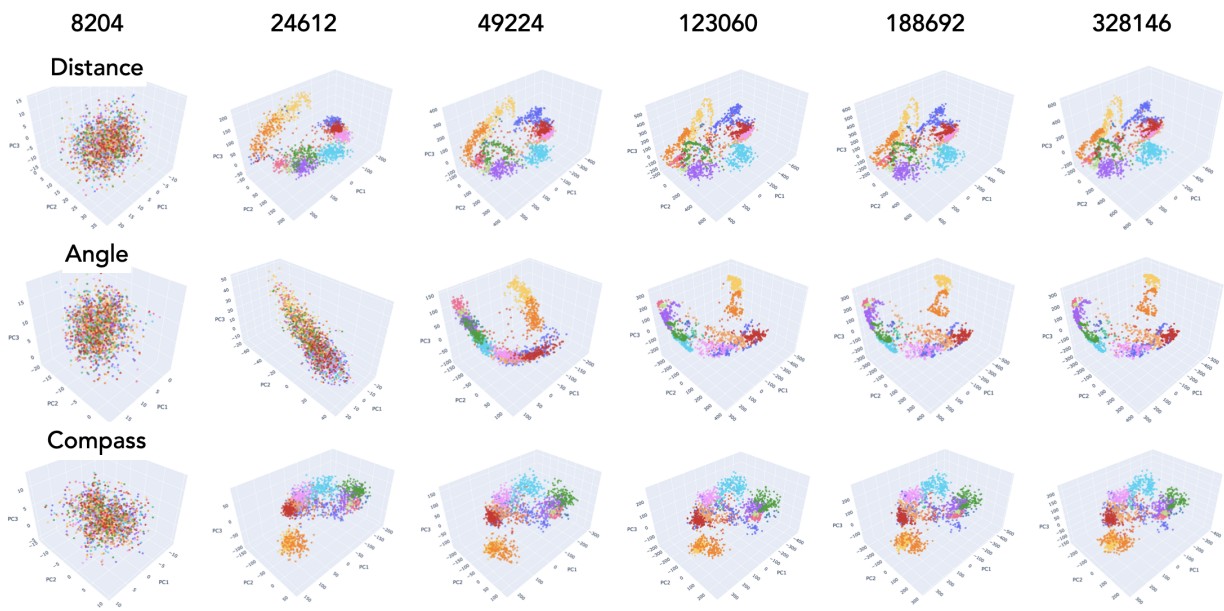

*Figure 10.* **Representation dynamics during training.** Rows: `distance` (top), `angle` (middle), `compass` (bottom). Columns show PCA projections at gradient steps 8204, 24612, 49224, 123060, 188692, and 328146 (left to right). Cities are colored by geographic region.

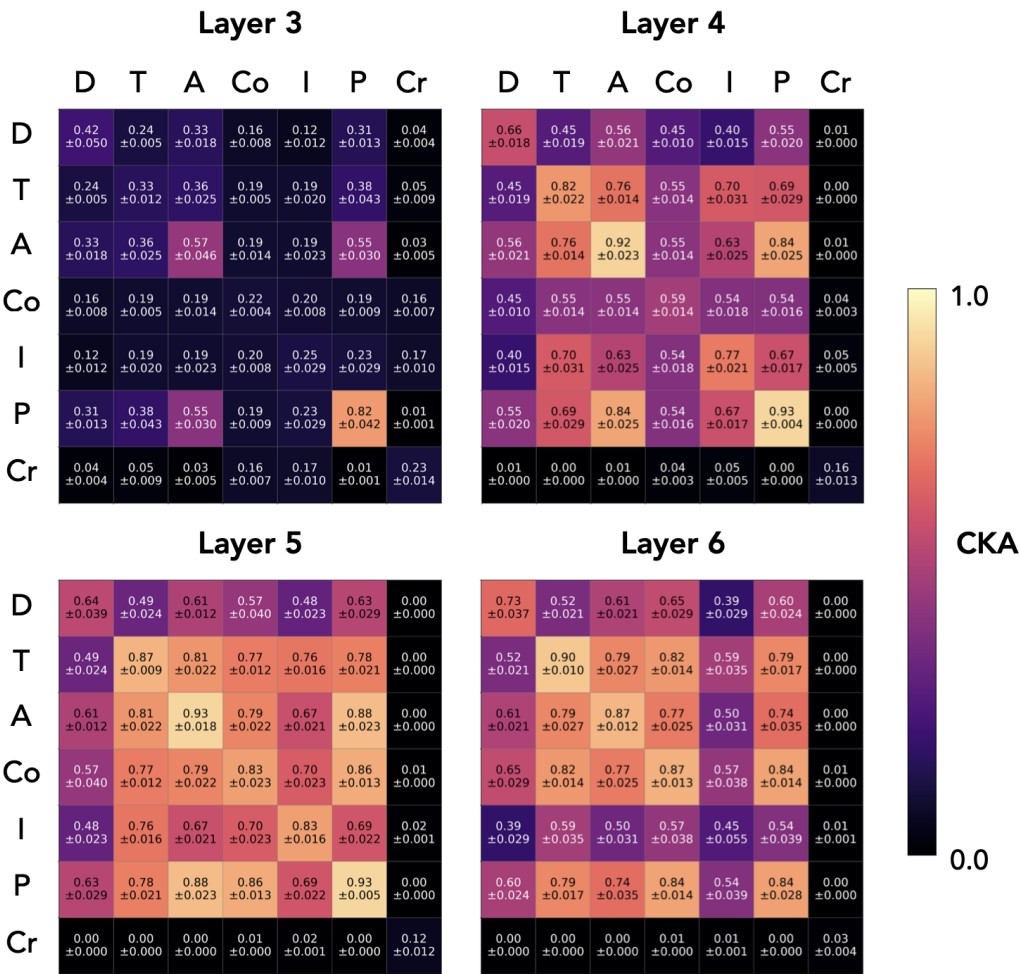

*Figure 11.* **CKA matrices for single-task models across layers.** Each cell shows mean ± SEM across 3 seeds. D=distance, T=triangle area, A=angle, Co=compass, I=inside, P=perimeter, Cr=crossing. CKA increases in later layers; `distance` shows consistently lower cross-task similarity.

minimum CKA of 0.89, indicating that multi-task training also reduces inter-seed variance. For diagonal entries, we exclude same-seed comparisons (which trivially yield 1.0) and report only the upper triangle since the matrix is symmetric. This confirms the main text finding that multi-task training drives representational convergence.

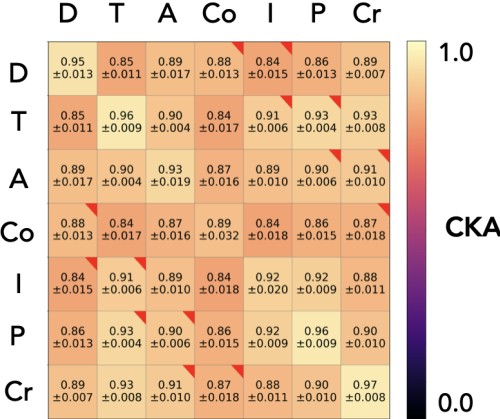

*Figure 12.* **CKA matrix for two-task models at layer 5.** Mean $\pm$ SEM across 3 seeds. All pairs show high alignment ($>0.84$), substantially higher than single-task models.

**CKA vs. Task Count (Per-Seed).** Fig. 13 shows the same CKA vs. task count analysis as Fig. 3(d) in the main text, but broken down by individual seeds. Each panel shows one seed. These per-seed values are pooled to produce the main text figure, where error bars represent SEM across seeds. The pattern is consistent across all three seeds: CKA increases substantially from 1 to 2 tasks and saturates at 2–3 tasks for layers 4–6.

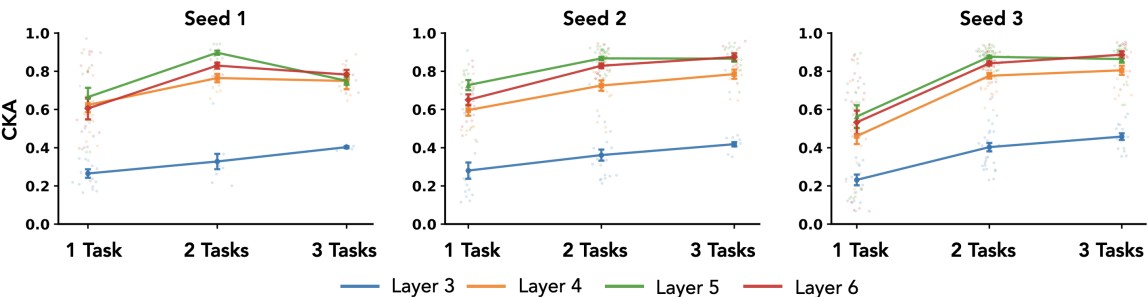

*Figure 13.* **CKA vs. task count for individual seeds.** Each panel shows a different seed. These values are pooled in Fig. 3(d); error bars there represent SEM across seeds.

**Aggregated CKA Trends.** Fig. 14(a) shows CKA vs. task count for a single seed, using all $\binom{7}{2} = 21$ two-task models and all $\binom{7}{3} = 35$ three-task models, but only comparing non-overlapping pairs (models sharing no common tasks). This yields 105 non-overlapping pairs for 2-task models and 70 for 3-task models. Fig. 14(b) shows within-task CKA (same task combination, different seeds) as a function of task count, demonstrating that multi-task training also reduces seed-to-seed variability: representations become more consistent not just across tasks but also across random initializations.

**CKA vs. Generalization (Annotated).** Fig. 15 is an annotated version of Fig. 5(b), with each point labeled by its (train→eval) task pair.

### D.4. Additional Fine-Tuning Evaluation Results

Raw fine-tuning results for individual seeds.

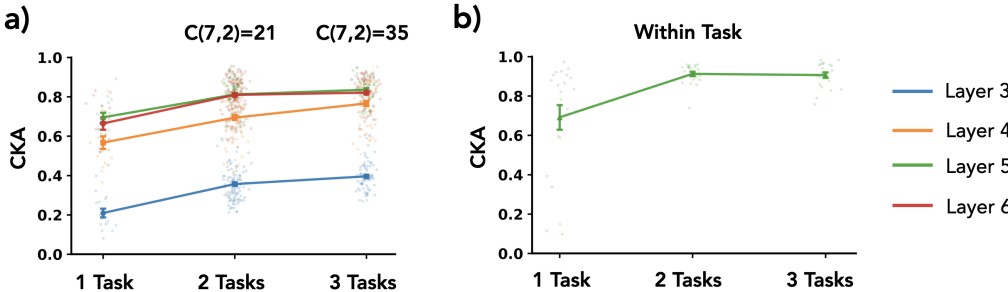

*Figure 14.* **Aggregated CKA analysis.** (a) CKA vs. task count for single seed, comparing only non-overlapping model pairs (105 pairs for 2-task, 70 pairs for 3-task). (b) Within-task CKA (same task combination, different seeds) increases with task count, indicating multi-task training reduces seed variability.

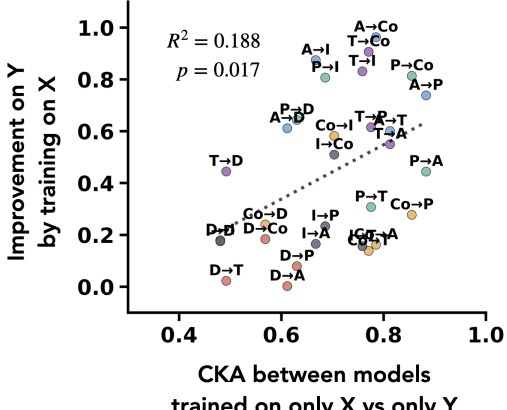

*Figure 15.* **Annotated version of Fig. 5(b).** Each point is labeled with its (train→eval) task pair. D=distance, T=triangle area, A=angle, Co=compass, I=inside, P=perimeter.

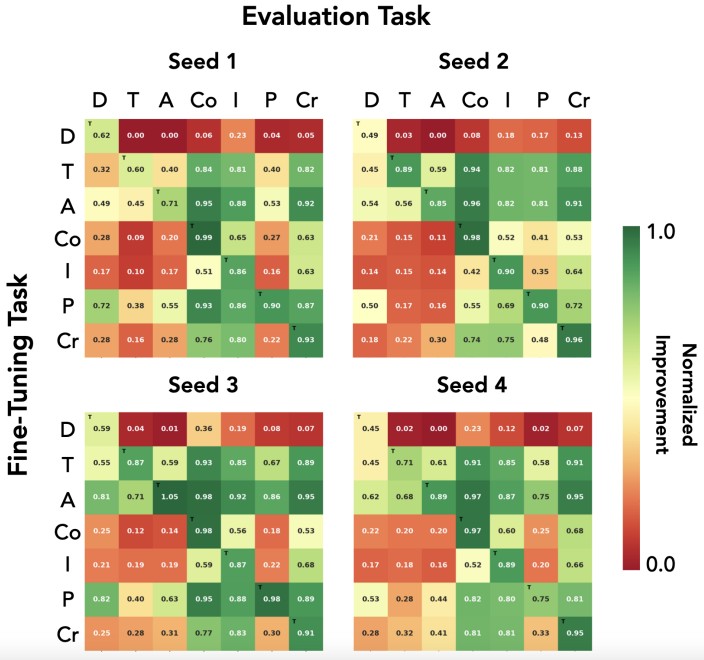

*Figure 16.* **Single-task fine-tuning results for individual seeds.** Per-seed version of Fig. 5(a), organized in a 2×2 grid.

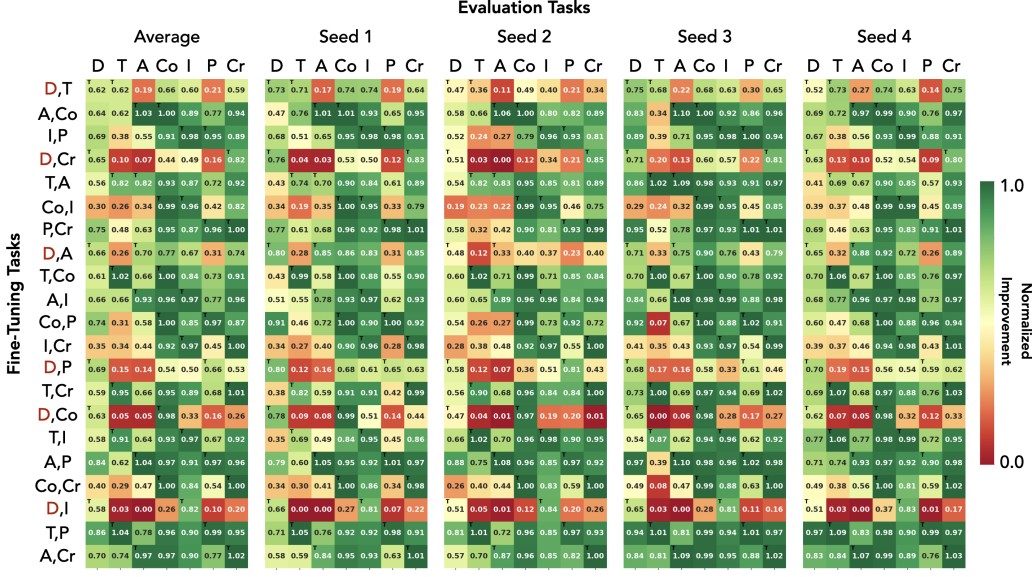

*Figure 17.* **Two-task fine-tuning normalized improvement for all 21 task combinations.** Leftmost panel shows average across seeds; remaining panels show individual seeds.

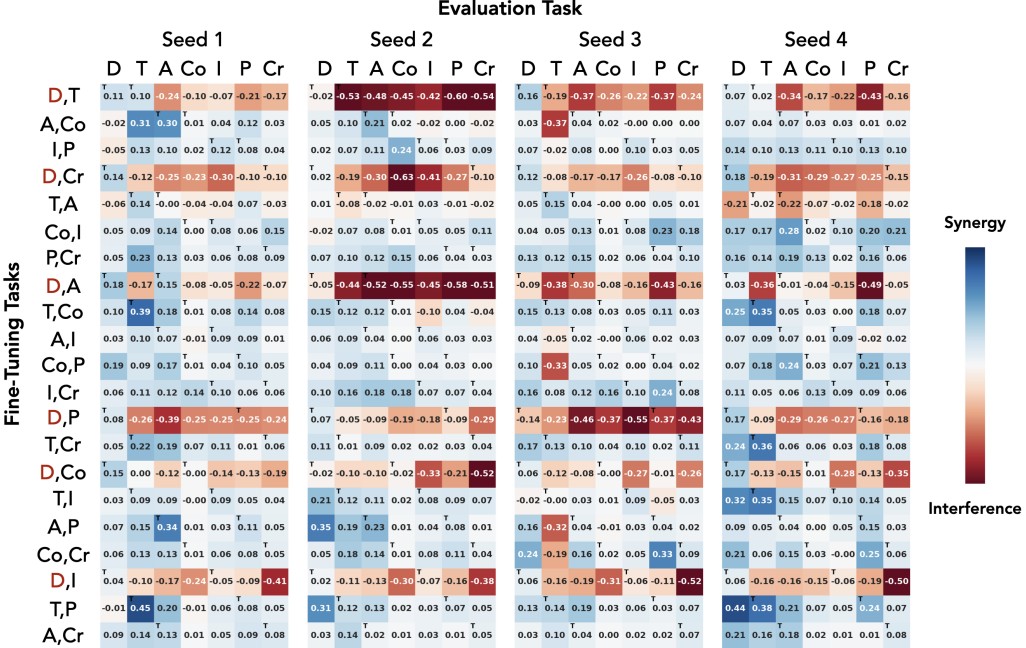

*Figure 18.* **Deviation from best-teacher expectation for all 21 two-task combinations.** All 4 seeds shown; average is in main text Fig. 6(c).

## D.5. Pretraining Variations

**Pretraining with `Atlantis`.** In the main text, we showed that fine-tuning on divergent tasks fails to integrate `Atlantis` cities into the learned representation manifold (Fig. 6d, red histogram). To verify that this failure stems from fine-tuning dynamics rather than a peculiarity of the geometry around `Atlantis`, we trained a model with `Atlantis` cities included from the start of pretraining. Fig. 19 shows the resulting representations: `Atlantis` cities are seamlessly integrated into the world manifold, indistinguishable from other cities in both PCA projections (a) and linear probe reconstructions (b). This confirms that the representation space can readily accommodate `Atlantis`, and thus, the integration failure observed in fine-tuning is a property of the optimization dynamics, not a fundamental limitation of the architecture or task.

**Wider Model.** To test whether our findings depend on model capacity, we trained a wider model with 2× the hidden dimension (256 vs. 128) and intermediate size (1024 vs. 512), resulting in approximately 4× the parameters. Fig. 20 shows fine-tuning results for this wider model: (a) single-task fine-tuning normalized improvement; (b) two-task fine-tuning normalized improvement; (c) deviation from best-teacher expectation. We still observe that `distance`-containing combinations (red labels in panel c) show degraded cross-task generalization. This suggests that divergent task interference is not simply a capacity limitation.

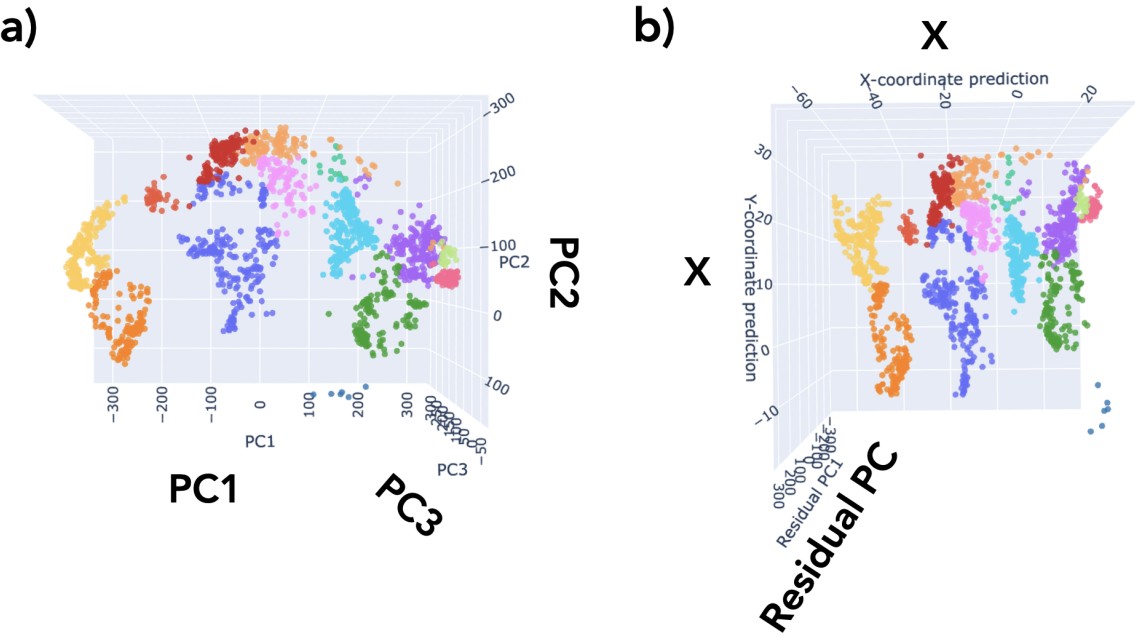

*Figure 19.* **Representations when `Atlantis` is included during pretraining.** (a) PCA projection showing `Atlantis` cities (small cluster in Atlantic region) integrated with world cities. (b) Linear probe reconstruction confirming geographic accuracy. Unlike fine-tuned models, `Atlantis` cities lie on the same manifold as other cities.

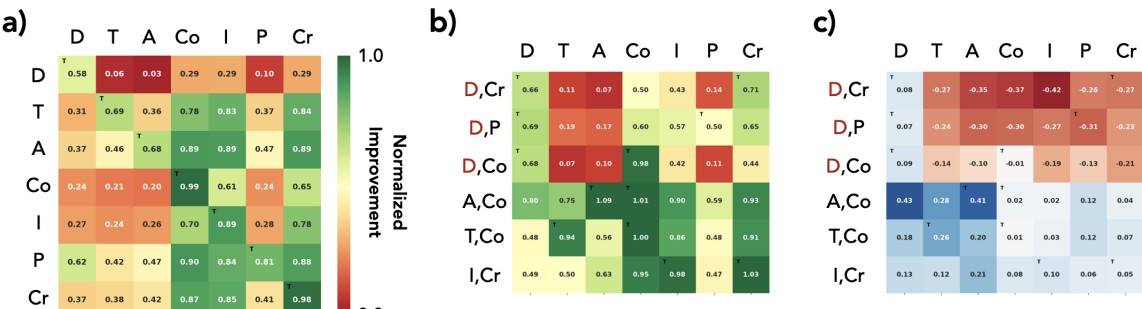

*Figure 20.* **Fine-tuning results for wider model (2× hidden dimension).** For all panels: rows = fine-tuning task(s), columns = evaluation task. (a) Single-task fine-tuning normalized improvement. (b) Two-task fine-tuning normalized improvement. (c) Deviation from best-teacher expectation; `distance`-containing combinations (red labels) still show degraded generalization.

