# OpenReview forum: "Convergent World Representations and Divergent Tasks"
_ICML.cc/2026/Conference — ICML 2026 regular_

### Official Review · Reviewer_CBid · 2026-03-08

**Soundness:** 3
**Presentation:** 3
**Significance:** 3
**Originality:** 3
**Overall Recommendation:** 4
**Confidence:** 3

**Summary:**

This paper studies how neural networks form internal world representations, when those representations converge across tasks, and whether fine-tuning can correctly integrate new entities into an existing representation space. It builds a synthetic setup by training an autoregressive Transformer on a world with 5,075 real city coordinates, and the training data comes from 7 geometric tasks defined over those cities, such as distance, angle, etc. One finding is that different singular tasks can produce different internal geometries even though they come from the same underlying world. Second, multi-task training encourages these representations to converge. Finally, the paper finds "divergent tasks," which hurt integrating new entities during fine-tuning and also harm cross-task generalization.

**Compliance With Llm Reviewing Policy:**

Affirmed.

**Final Justification:**

The rebuttal reinforced my prior assessment that this is a well-executed paper with interesting insights. I think the main concern is that there is a lack of mechanistic account and this is still a toy setting. The authors did provide some attempts at a mechanistic account but it's not causal or predictive. This is the main reason that I didn't raise my score, but I still think it's a good paper.

**Key Questions For Authors:**

**Weaknesses**
* Results 1 and 2 seems well-established by works such as [1,2] mentioned above. Are these claims meant to be a novel contribution or is meant to be a corroboration of prior results in this paper's set-up?
* How does representational alignment and/or generalization vary with pre-training and/or fine-tuning data scale?
* Can the authors mechanistically demonstrate how divergent tasks induce gradient updates that bypass shared representations and harm integration of new entities?

**Limitations:**

Yes.

**Strengths And Weaknesses:**

**Strengths**
* The biggest strength is the very clean, creative experimental framework of the 5,075 real city coordinates and multiple geometric tasks. The model does not have direct access to the latent states, which allows later probing of the model for its internal "world model" and if it recovers these states.
* The paper is generally well-written and has nice visuals that accompany the arguments.
* The main surprising claim is that representation convergence during pre-training does not guarantee good representational integration during fine-tuning. The paper posits an interesting notion of "forward and backward-pass modularity," where a model can appear to use a shared representation at inference time, yet gradients from a task may not update that shared space in a way that supports broad generalization.

**Weaknesses**
* Many of the findings align closely with existing literature. In fact, many of the references in this paper directly point to these claims/findings. "Result 1: World Representations Emerge through Autoregressive Training" (L205-207) has been shown in [1]. "Result 2: Divergent Tasks Catastrophically Harm Generalization" (L344-346) is supported by prior work like [2] which shows how for large distribution shifts, fine-tuning can distort pre-trained features. This paper already cites [1, 2]. So, the main novelty appears to lie somewhat on the finding of entity integration and the controlled synthetic framework.
* The paper makes claims about how divergent tasks induce gradient updates that bypass shared representations and harm integration of new entities. However, these claims are mostly speculative/correlational rather than mechanistically demonstrated.
* The paper does not study how representational alignment or generalization varies with dataset scale. The experiments use a fixed pre-training budget and a fixed fine-tuning recipe, so it is unclear whether the reported “divergent task” effects reflect a robust representational phenomenon or partly differences in data efficiency across tasks. A scaling analysis over pre-training and/or fine-tuning data would strengthen the claim.

[1] Li et al. "Emergent world representations: Exploring a sequence model trained on a synthetic task." ICLR 2022.

[2] Kumar et al. "Fine-tuning can distort pretrained features and underperform out-of-distribution." ICLR 2022.

---

> ### Author Rebuttal · Authors · 2026-03-31
>
> ## **Response to Reviewer CBid**
>
> Thank you for the thorough review! We are glad you found the framework **"very clean"** and **"creative"** and recognized the forward/backward-pass modularity as **"surprising."**
>
> We ran some additional experiments to improve the paper. In specific we did:
>
> 1. 4 additional representation metrics (CKA, RSA, SNN, SVCCA) on all 1,2,3 task pretrained models.
> 2. An effective rank analysis.
> 3. Scattered Atlantis experiment (1pretraining + (21+7) fine tunings)
> 4. 2x width ablation.
>
> Please find the plots supporting the below at [**[Supplementary Page](https://anonymous.4open.science/r/anonymous-icml-rebuttal-crdt-FE8D/README.md)**].
>
> ---
>
> **W1: Novelty relative to Li et al. and Kumar et al.**
>
> *Li et al. (2022)*: We totally agree that "representations emerge" alone is not new at all. Our contribution is studying multiple tasks on the *same* world and showing that despite they individually produce different geometries multi-task training drives representational convergence. PRH (Huh et al., 2024) explicitly hypothesizes that multi task learning constrain the supporting representation space and we think we might be among the first to explicitly demonstrate this in a controlled setup.
>
> *Kumar et al. (2022)*: While Kumar et al. is superficially related in studying fine-tuning, we do not think what they find is similar to our findings. Kumar et al. shows that fine-tuning *generally* distorts features during optimization. In our setting, fine-tuning harmlessly integrates Atlantis for 6/7 tasks while the distance task actively harms integration of *new* entities. Moreover, we believe the main defining factors of our setup: pretraining world, new entities, compositional entity-task setup which allows the study of continual learning differentiates from Kumar 2022. Furthermore, Kumar 2022 does not explicitly perform a representational analysis.
>
> ---
>
> **W2: Speculative mechanism.** This is a fair critique, we do note in the manuscript that *"We clarify that our findings are correlational"* and *"We identify divergence as a diagnostic marker but do not reveal underlying mechanisms."* Overall we agree the representational findings are correlational. However, we do think forward/backward modularity can be well defined without a mechanistic understanding, purely by measuring generalization performance.
>
> Nevertheless, we strengthened the robustness of our phenomenology by:
> 1. 4 seeds (App. D)
> 2. 2 model width (Supp. Fig. 4)
> 3. Training on a world where Atlantis is scattered around the world (Supp. Fig. 3a, b)
>
> ---
>
> **W3/Q2: No scaling analysis.** Our intuition was that this is not solved by scale, since it stems from representational geometry rather than capacity. To confirm, we trained a 2x wider model (256 hidden, 1024 intermediate, 8 heads) on the same data and same epoch and ran the same fine-tuning experiments. **We find the divergent task pattern is unchanged** (Supp. Fig. 4), where distance task is still the harming one.
>
> ---
>
> **Q1: Novel contribution or corroboration?** We think section 4 result 1 is not novel (l.195: "~the emergence of linearly decodable coordinates might be anticipated"), while **section 4 result 2, 3 are genuinely novel results**: "different tasks produce different geometries" and "multi-task training drives representational convergence". **All results in section 5 seem novel to the best of our knowledge** while *similar* phenomena has been observed (Kumar 2022).
>
> (We will renumber the result numberings, sorry for the confusion.)
>
> ---
>
> **Q3: Mechanistic demonstration.** We agree this is missing. We have converging empirical evidence (see W2) but not a mechanistic account.
>
> ---
> ---
>
> **Summary:** We thank the reviewer again for the thorough and constructive feedback! We believe the framing of the paper greatly improved thanks to the reviewer's comments. We hope these additions address your concerns and we would be grateful if you would consider recommending our paper more strongly.
>
> We are happy to answer any follow up questions!

---

> > ### Author Rebuttal · Reviewer_CBid · 2026-04-02
> >
> > Thanks for clarifying my questions, especially regarding the points about novelty! My concerns are partially resolved, the main  one left is the lack of a causal/mechanistic account, which the authors acknowledge. The authors have robustified their claims with more experiments which strengthens the evidence. I think this is pretty good paper in its current form, and the lack of any causal or theoretical perspectives is something that will be difficult to resolve in a short rebuttal period.
> >
> > On another note, thanks for running the experiment with a 2x wider model! But by a "scaling analysis over pre-training and/or fine-tuning data," I meant running a sweep over the amount of pre-training/fine-tuning data to see if the phenomena being observed is due to data efficiency across tasks. That said, I don't think this a huge deal that would affect my score.

---

> > > ### Author Response · Authors · 2026-04-04
> > >
> > > Thanks for pushing us on the mechanistic question!
> > >
> > > Motivated by your comment, we ran additional experiments directly analyzing the gradients from the different fine-tuning tasks. We got some interesting mechanistic insights:
> > >
> > >  - **What we did:** For each of the 7 single-task fine-tuning objectives, we computed gradients at two levels: (1) w.r.t. all model parameters (the standard parameter-space view, ~640 gradient samples per task), and (2) w.r.t. hidden activations at the novel entity's token position (the representation-space view, ~5,500 gradient samples per task). We then measured pairwise cosine similarity across all 7 tasks at both levels. All at layers 3-5.
> > > - **Finding 1 [Supp. Fig. 5a]:** Parameter-space gradients are uninformative: All task pairs show near-zero cosine similarity (0.04-0.25), with no task standing out as an outlier. This is unsurprising in high dimensions, where random vectors are approximately orthogonal. The most negative pair is actually
> > >   inside-distance (-0.32), not distance alone. At the parameter level, the tasks are largely indifferent to each other (Supp. Fig. 5a).
> > > - **Finding 2 [Supp. Fig. 5b,c]:** But in the activation space at Atlantis positions, distance is the clear outlier: When we look at what those parameter updates do to the novel entity's internal representation, the picture changes completely. The six non-distance tasks are strongly mutually aligned (cossim 0.60-0.95), all agreeing on which direction to push the entity's hidden state. Distance is anti-correlated with every single one of them at every layer (Supp. Fig. 5b). Projecting the mean activation gradient onto the coordinate probe subspace (Supp. Fig. 5c), distance is the only task with a negative X-projection: it pushes new entities in the opposite direction along the primary coordinate axis.
> > >
> > > ---
> > >
> > > **This means:** Six tasks agree: "push the new entity's representation this way to integrate it into the coordinate system." Distance says: "push it the opposite way." This seems like a concrete, representation-level conflict, strongest at layers 3-4, which our intervention analysis independently identified as the causal computation layers.
> > >
> > > ---
> > >
> > > We think this gives at least a first mechanistic account of why divergent tasks harm entity integration! We wanted to share these interesting results whether or not that affects the score!
> > >
> > > All plots are in Section 5 of the same link [**[Supplementary Page](https://anonymous.4open.science/r/anonymous-icml-rebuttal-crdt-FE8D/README.md)**]
> > >
> > > Thank you again for raising this point!
> > >
> > > ---
> > > p.s. We also added some experiments where we withhold different regions instead of Atlantis during pretraining (Supp. Section 6). All findings reproduce

---

### Official Review · Reviewer_rQVL · 2026-03-10

**Soundness:** 2
**Presentation:** 3
**Significance:** 3
**Originality:** 3
**Overall Recommendation:** 4
**Confidence:** 3

**Summary:**

The paper builds a synthetic “world” consisting of 5,075 real city coordinates together with 100 synthetic “Atlantis” cities. From this world, the authors define seven geometric tasks (distance, triangle area, angle, compass, inside, perimeter, and crossing) to generate supervised training data. Autoregressive transformers are then trained from scratch on individual tasks as well as on multi-task combinations.

**Compliance With Llm Reviewing Policy:**

Affirmed.

**Key Questions For Authors:**

1. Have you experimented with adding textual or sequential tasks (for example next-token prediction) to test whether the same convergence behavior appears?
2. Could a curriculum strategy, where divergent tasks are introduced later in training, reduce the fine-tuning failure?
3. How sensitive are the results to the temperature parameter γ in the softmax similarity encoding?
4. Would increasing the codebook size for the line-manifold quantizer change the task-specific geometry that emerges?

**Limitations:**

yes

**Strengths And Weaknesses:**

Strengths: A carefully constructed synthetic benchmark with known ground-truth geometry, allowing detailed analysis of representation structure. Empirical results show that multi-task training tends to align internal representations across tasks (supported by CKA and PCA analysis). The paper includes several ablation studies across task counts, random seeds, and model width, along with visualizations of representation behavior.

Weaknesses: The experiments use relatively small transformer models (≤256 hidden dimension) and synthetic geometric tasks, so it is unclear whether the findings transfer to large language models or vision systems. Evaluation focuses mainly on geometric metrics (e.g., coordinate reconstruction); no semantic or linguistic downstream tasks are considered. Although CKA is used extensively, additional similarity metrics (e.g., SVCCA or RSA) could provide further insight.

---

> ### Author Rebuttal · Authors · 2026-03-31
>
> ## **Response to Reviewer rQVL**
>
> Thank you for the review! We appreciate your recognition of the **"carefully constructed synthetic benchmark"** and **"several ablation studies."**
>
> We ran some additional experiments to improve the paper. In specific we did:
>
> 1. 4 additional representation metrics (CKA, RSA, SNN, SVCCA) on all 1,2,3 task pretrained models.
> 2. An effective rank analysis.
> 3. Scattered Atlantis experiment (1pretraining + (21+7) fine tunings)
> 4. 2x width ablation.
>
> Please find the plots supporting the below at [**[Supplementary Page](https://anonymous.4open.science/r/anonymous-icml-rebuttal-crdt-FE8D/README.md)**].
>
> ---
>
> **W1: Small models, unclear transfer.** Here, we intentionally keep the model small to enable exhaustive experimentation (during the research process we ran 272+ models across seeds, tasks, architectures). At the same time, we designed the setup to be as realistic as possible within this constraint: rotary positional embeddings, multi-token entity IDs (not one token per city), character-level numerical outputs requiring compositional computation, etc. Motivated by the reviewer's comment we experimented with a model with 2x width (256 hidden, 1024 intermediate, 8 heads; Supp. Fig. 4). Our findings on the divergent task is shown to persists at larger scale. However, we totally agree additional modalities will be a rich future study.
>
> ---
>
> **W2: Only geometric evaluation.** The geometric ground truth (known city coordinates) is what makes our analysis of "world" representations possible. In settings with semantic or linguistic structure, there often lacks a verifiable structure one could expect the model to internalize and thus making it hard to draw precise conclusions about representation geometry. However the geometric setting allows us to comapre the model's internal geometry to the city coordinates' geometry.
>
> ---
>
> **W3: Only CKA. [Supp. Sec. 1]** We computed four metrics: CKA, RSA, SNN (k=10), SVCCA across all pairwise comparisons (10,080 computations). Cross-task similarity between models with no shared tasks at layer 5:
>
> | Metric | 1 task | 2 tasks | 3 tasks |
> |--------|:---:|:---:|:---:|
> | CKA | 0.501 | 0.879 | 0.854 |
> | RSA | 0.467 | 0.892 | 0.885 |
> | SNN | 0.101 | 0.185 | 0.204 |
> | SVCCA | 0.551 | 0.729 | 0.794 |
>
> All confirm convergence, consistent across layers 3-5 (Supp. Fig. 1e). SNN (the non-global metric from the PRH paper) supports it as well (despite SNN values are smaller by construction). SVCCA (subspace alignment; Kornblith et al. 2019) shows alignment even across single-task models, while CKA/RSA/SNN capture finer geometric differences that converge with multi-task training. We note that while PRH focused on SNN, we do not consider SNN to be, in any sense, a "superior" metric compared to CKA.
>
> ---
>
> **Q1: Textual/sequential tasks.** We want to make sure we address the right concern. Our training is standard autoregressive next-token prediction where the model predicts character sequences like "d i s t ( c _ 0 8 6 5 , c _ 4 8 7 9 ) = 7 6 9" token by token, with no supervision beyond cross-entropy loss. If the reviewer is asking how tasks with richer linguistic structure affect the framework, that is a great question but is probably beyond the score of this work!
>
> ---
>
> **Q2: Curriculum for divergent tasks.** We did not focus on this, but our pretraining-with-Atlantis experiment is relevant: when Atlantis is included from pretraining, integration is perfect (probe error matches regular cities). This means that at least there is *some* curriculum which determines whether new entities can be well integrated or not. To control for purely geometrical causes we also ran a scattered Atlantis experiment (100 cities uniformly distributed, 1+21+7 new models) confirming the divergent task effect is not location-dependent (p=0.0001, Supp. Fig. 3a-b). Thus we conclude the iiming of entity introduction clearly matters, and curriculum-based approaches are a promising direction.
>
> ---
>
> **Q3: Temperature.** We are not sure what this refers to. Our framework does not include a temperature parameter γ in a softmax similarity encoding. We performed all evaluation is at temperature 0.0 (greedy decoding). We did an early evaluation at temperature 1.0 for generation, but the results were mostly unchanged.
>
> ---
>
> **Q4: Codebook size.** We are a bit confused we do not have a codebook for a line-manifold quantizer. Would you be able to elaborate this question?
>
> ---
> ---
>
> **Summary:** We thank the reviewer again for the constructive feedback! We hope these additional experiments and clarifications address your concerns and we would be grateful if you would consider recommending our paper more strongly.
>
> We are happy to answer any follow up questions!

---

> > ### Author Rebuttal · Reviewer_rQVL · 2026-04-05
> >
> > The rebuttal has helped clarify several points, but it does not sufficiently address all of my concerns. Therefore, I will maintain my original score. in particular, I still have follow-up questions concerning the robustness of the method under varying conditions and the justification of certain design choices. I encourage the authors to further elaborate on these aspects to strengthen the overall contribution and improve the clarity and rigor of the work.

---

> > > ### Author Response · Authors · 2026-04-05
> > >
> > > Dear Reviewer rQVL,
> > >
> > > Thank you for your acknowledgement and your time throughout this revision process!
> > >
> > > You mention still having "follow-up questions concerning the robustness of the method under varying conditions and the justification of certain design choices." Could you please state these explicitly so we can address them?
> > >
> > > For context, here is a summary of all the additional experiments we have run across our rebuttals:
> > >
> > > - 4 new representation metrics (CKA, RSA, SNN, SVCCA) across all model combinations
> > > - Effective rank analysis for all 63 models
> > > - 2x width ablation confirming divergent task effect at larger scale
> > > - Scattered Atlantis controls (22 new models) ruling out location dependence
> > > - Gradient-level mechanistic analysis (Supp. Sec. 5) showing distance produces anti-correlated activation gradients at novel entity positions while the other 6 tasks are mutually aligned
> > > - 87 additional models holding out 3 geographic regions during pretraining (Supp. Sec. 6), confirming distance remains divergent regardless of holdout region
> > >
> > > These are available at the same link: [[**Supplementary Page**](https://anonymous.4open.science/r/anonymous-icml-rebuttal-crdt-FE8D/README.md)]
> > >
> > > Given these, could you clarify:
> > >
> > > 1. Which specific concerns remain unresolved?
> > > 2. Does the above experiments (spanning different data splits, model sizes, metrics, and a mechanistic study) cover what you mean by "robustness under varying conditions" and "justification of design choices"?
> > >
> > > We would like to make sure we have a fair chance to address the remaining questions before the discussion period ends.
> > >
> > > Thank you very much for your time!

---

### Official Review · Reviewer_pKZe · 2026-03-11

**Soundness:** 3
**Presentation:** 3
**Significance:** 2
**Originality:** 3
**Overall Recommendation:** 5
**Confidence:** 3

**Summary:**

The paper studies the interplay between training tasks and neural representation geometry. Based on an experimental suite consisting of multiple geometric tasks on a 2D world map dataset, the paper provides evidence of multi-task training driving representation convergence. Additionally, through the lens of ‘novel entity’ adaptation, the authors reveal the existence of divergent task(s) characterized as those that harm multi-task generalization performance when incorporated.

**Compliance With Llm Reviewing Policy:**

Affirmed.

**Final Justification:**

The paper is a well conducted empirical study providing relevant insights on how learning on different tasks affects the representations learned. However, in my view, the main limitation of the work is that it is unclear how the analysis plays out in more complex and realistic settings.

During the rebuttal the authors conducted additional experiments providing a more rounded evaluation on the synthetic setting. I still have some reservations wrt. the geographic holdout setting where only few relatively clustered locations were removed (North Africa, Middle east. North India), raising questions on whether these were truly random (i.e., randomized analysis Q2a) or cherry picked. In regards to this, I would have expected a more diverse setup where the pretraining world varies significantly in convex hull (i.e., shape of the world).  I will give the authors the benefit of the doubt but kindly ask them to elaborate why only these regions were considered in the revised version.

Overall, in my opinion, the paper lays a solid foundation for other works to expand on, in which case, a purely synthetic but controllable setting is a somewhat defensible choice. Finally, acting in good faith, the rebuttal changed my evaluation and I accordingly raise my recommendation score by 1.

**Key Questions For Authors:**

(i) According to Fig 2. b. only the “Crossing” fails to train in isolation.  (ii) According to Fig 6 a. only including the “Distance” appears to harm the fine tuning integration. Based on these, (i) is attributed to hard-to-learn dynamics (footnote 2). On the other hand, the “Distance” negatively impacting integration is framed as a fundamental property indicating the existence of divergent task(s).

* Q1. How can one be certain that the failures under (i) and (ii) are not connected? In principle one could also claim that the interference between “Distance” and the other tasks is not fundamental but rather a byproduct of the optimization process (i.e., similar to footnote 2).

&nbsp;

The fine tuning integration setting was restricted to “Atlantis” entities. I found this limiting in two aspects: (i) new entities from “Atlantis” only probe the behavior on this specific part of the world model and (ii) new entities are only locally distributed.

* Q2a. How can one be certain that the effects observed are universal (within the synthetic setting). For example, it could be the case that some other tasks are divergent when “Africa” is held out and “Atlantis” included during the pretraining.
* Q2b. A natural extension to the above, does the finding remain intact when the new entities are distributed? (e.g., new entities originate from either “Atlantis1” and “Atlantis2” placed in the Atlantic and Pacific oceans respectively).

&nbsp;

* Q3. When quantifying the representation alignment, have you considered any other similarity metrics apart from CKA?

I think addressing Q3. will not only provide a more complete picture but also strengthen the connecttion to the original RPH paper [1]. Recall that [1] reported both global and non-global metrics to characterize the representation alignment where they found that the shared nearest neighbor metric (non-global) provides stronger support for the hypothesis compared to CKA (global).

* Other points for your consideration:

    - In relation to Sec 4. Results 2, although qualitative inspection offers some high-level insights a more thorough analysis of the emerged representation geometries is required to better understand their nature. For example one could consider reporting some quantitative metrics to characterize the geometries e.g., representation rank [2]?

    - Fig 5a. is impossible to read, either remove the numbers inside the cells and provide a larger version in the supp or increase the font.

    - In Fig 5b. the horizontal and vertical axes express symmetric and non-symmetric quantities which I found to be somewhat counterintuitive when interpreting their correlation.

[1] Huh, Minyoung, et al. "Position: The platonic representation hypothesis." Forty-first International Conference on Machine Learning. 2024.

[2]  Masarczyk, Wojciech, et al. "The tunnel effect: Building data representations in deep neural networks." Advances in Neural Information Processing Systems 36 (2023): 76772-76805.

**Limitations:**

Yes.

**Strengths And Weaknesses:**

**Strengths:**

* S1. The paper studies the representation geometry which is a fundamental problem to deep learning with relevant applications such as ML-interpretability
* S2. The paper, for the most part, is well written and clearly communicates its findings.
* S3. The paper reveals interesting insights on the interplay between multi-task training and the emerging neural representations. Additionally, the findings connect to a trending hypothesis (i.e., Platonic Representation Hypothesis) and therefore are likely to be of interest to many in the community.

&nbsp;

**Weaknesses:**

* W1a. The experiments are conducted solely on synthetic data raising questions on whether the conclusions are universal across more complex/realistic settings and architectures.
* W1b. Even under the 2D synthetic world data restriction, I found the experimental analysis narrow in some aspects.
* W2. Although interesting observations were made in the paper, limited effort was made in understanding their underlying nature.

---

> ### Author Rebuttal · Authors · 2026-03-31
>
> ## **Response to Reviewer pKZe**
>
> Thank you for the detailed review! We are glad you found the paper **"well written"**, our insights on multi-task training **"interesting"**, and the PRH connection **"likely to be of interest to many in the community."** Your suggestions directly motivated substantial new experiments. In specific we did:
>
> 1. 4 additional representation metrics (CKA, RSA, SNN, SVCCA) on all 1,2,3 task pretrained models.
> 2. An effective rank analysis.
> 3. Scattered Atlantis experiment (1pretraining + (21+7) fine tunings)
> 4. 2x width ablation.
>
> Please find the plots supporting the below at [**[Supplementary Page](https://anonymous.4open.science/r/anonymous-icml-rebuttal-crdt-FE8D/README.md)**].
>
> ---
>
> **W1a: Synthetic data.** We agree. The controlled setting lets us make causal claims that observational studies cannot. The phenomena we observe do parallel real LMs (reversal curse, Berglund et al. 2024; knowledge editing failures, Lampinen 2025 ICL underperforming FT).
>
> ---
>
> **W1b\Q3: Narrow analysis.** We broadened our analysis with: (1) four metrics beyond CKA (Supp. Sec. 1, Fig. 1e), (2) effective rank for all 63 models (Supp. Fig. 2a-b), (3) scattered Atlantis, 22 new models (Q2b, Supp. Fig. 3a-b), (4) 2x width ablation (Supp. Fig. 4). Findings were largely consistent. Highlighting the diverse metrics, CKA, RSA, SNN (k=10), SVCCA across all pairwise comparisons gave cross-task similarity (no shared tasks), layer 5:
>
> | Metric | 1 task | 2 tasks | 3 tasks |
> |--------|:---:|:---:|:---:|
> | CKA | 0.501 | 0.879 | 0.854 |
> | RSA | 0.467 | 0.892 | 0.885 |
> | SNN | 0.101 | 0.185 | 0.204 |
> | SVCCA | 0.551 | 0.729 | 0.794 |
>
> All four metrics confirm convergence, consistent across layers 3-5 (Supp. Fig. 1e).
>
> ---
>
> **W2: Understanding the underlying nature.** This is a fair critique, we do note in the manuscript that "We clarify that our findings are correlational" and "We identify divergence as a diagnostic marker but do not reveal underlying mechanisms." Overall we agree the representational findings are correlational. However, we do think forward/backward modularity can be well defined without a mechanistic understanding, purely by measuring generalization performance.
>
> ---
>
> **Q1: Are crossing failure and distance divergence connected?** These are totally different phenomena. The crossing task, when trained in isolation simply gets stuck in a loss plateau where training loss stays flat, task accuracy at chance, no world model forms (effective rank ~161, Supp. Fig. 2). This is a known issue (Pezeshki et al., 2021; Hoffmann et al., 2024; Gopalani & Hu, 2025) that disappears when any companion task is added. However, crossing does *not* harm integration when included in fine-tuning.
>
> Distance trains successfully in isolation: loss plateau drops, prediction error is low, structured representations form (effective rank ~20, linearly decodable coordinates). But its geometry is distinct, and this mismatch harms integration of new entities when fine tuning on it. If this were "just optimization" like crossing, we would expect sensitivity to perturbations. Instead it persists across 4 seeds, 2x model width (Supp. Fig. 4), and scattered Atlantis (Supp. Fig. 3a-b). Crossing is fragile (any companion fixes it); distance's divergence is robust.
>
> ---
>
> **Q2a/Q2b: Universality and distributed entities.** We scattered 100 Atlantis cities uniformly across the globe and trained PT1 + all 21 fine-tuning combinations from scratch (22 new models). Finding stay robust: Distance still harms integration (Supp. Fig. 3a-b). On the other hand, when Atlantis is included from pretraining, integration is perfect, thus confirming the effect is about fine-tuning dynamics.
>
> ---
>
> **Q3:** See W1b above. We generally find convergence across all representational metrics, despite strongest in CKA
>
> ---
>
> **Representation rank. [Supp. Fig. 2a-b]** Following your suggestion and [2], we computed effective rank (participation ratio) for all 63 experiments, layers 3-6, we find, as expected that fuzzy representations have a much higher rank (100+ instead of 15~30).
>
> ---
>
> **Fig. 5a, b:** Thank you for these comments, we enlarged Fig. 5a and made the caption of Fig. 5b more clear by clarifying the mapping from Fig. 5a to 5b in the caption: each off-diagonal cell in 5a becomes one point in 5b, with the corresponding single-task CKA on the x-axis.
>
> ---
> ---
>
> **Summary:** We thank the reviewer again for the detailed and constructive feedback. Your suggestions for additional metrics, effective rank, and scattered Atlantis directly motivated experiments that we think substantially strengthen the paper. We hope these additions address your concerns and we would be grateful if you would consider recommending our paper more strongly.
>
> We are happy to answer any follow up questions!

---

> > ### Author Rebuttal · Reviewer_pKZe · 2026-04-02
> >
> > Thank you for the response. The additional results complement the original evidence further increasing the soundness of the setup. My remaining reservations are wrt W1a and Q2.
> >
> > When it comes to W1a I understand that extending the analysis on real data comes with less design control and is non-trivial, which nevertheless remains the main limitation of the paper.
> >
> > Your scattered Atlantis is essentially a limiting case of Q2b and rules out the possibility that the observed phenomena is dependent on the geometry of the fine-tuning split (as the same behavior emerges both when Atlantis is clustered or scattered). However, it does not account for Q2a which would have ruled out the possibility that the observed phenomena interacts with the geometry of the pretraining data.
> >
> > If the authors can provide a convincing argument on why Q2a is trivial (not worth answering) or alternatively a randomized analysis addressing Q2a I will increase my score by 1.

---

> > > ### Author Response · Authors · 2026-04-04
> > >
> > > Thank you for pointing this one out, indeed we kind of missed Q2a ! We now ran 87 additional models to directly address Q2a.
> > >
> > > **New experiment (Supp. Sec. 6, Figs. 6a-e):** We held out three different geographic regions during pretraining: North Africa (234 cities), North India (259 cities), and Middle East (210 cities), and ran the full 7task pretraining, 1 task FT and 2 task FT for each (1+7+21 models per region).
> > >
> > > **All findings reproduce across all three holdout regions:**
> > >
> > > - **1 task FT (Fig. 6b):** Distance remains the worst single-task specialist at transferring to other tasks (avg transfer: NA=0.11, NI=0.12, ME=0.04), while other tasks average 0.43-0.50. No other task becomes divergent when the holdout region changes.
> > > - **2 task FT - best-teacher (Fig. 6c):** All 6 distance-containing pairs show interference (red rows), while non-distance pairs are neutral to mildly synergistic. The pattern is strikingly consistent across all three regions.
> > > - **Probe generalization (Figs. 6d-e):** We also extracted representations and trained linear probes to predict coordinates. A non-distance model (angle+perimeter) integrates holdout cities at baseline accuracy (error 112-128 vs baseline 119-133). A distance-containing model (distance+inside) shows 3-4x worse probe error (318-435), confirming that the harm extends to the representational level.
> > >
> > > This rules out Q2a: distance divergence is not an artifact of which region is held out. Combined with the scattered Atlantis experiment (Sec. 3, Q2b), seed robustness (4 seeds, App. D), 2x width (Sec. 4), and the gradient analysis (Sec. 5), the divergent task phenomenon is thoroughly established as a property of the task itself.
> > >
> > > All plots are at the same link! [[**Supplementary Page**](https://anonymous.4open.science/r/anonymous-icml-rebuttal-crdt-FE8D/README.md)], Section 6.
> > >
> > > We hope this addresses the remaining concern!
> > >
> > > ---
> > >
> > > p.s. We also ran some mechanistic studies (Supp. Section 5) showing that distance task doesn't have lower or negative parameter space gradients, *but* the gradients on atlantis' representation is indeed very different from the 6 other tasks.

---

### Official Review · Reviewer_hkRS · 2026-03-12

**Soundness:** 3
**Presentation:** 4
**Significance:** 3
**Originality:** 4
**Overall Recommendation:** 5
**Confidence:** 3

**Summary:**

The authors focus on a fundamental problem: how world representations emerge, converge, and adapt during autoregressive training and fine-tuning. This paper studies world models in a highly controlled synthetic setting built from real-world cities, where each city is associated with hidden geographic coordinates. Models are trained with next-token prediction on tasks involving spatial relations among cities, such as distance, angle, and geometric inclusion, in both single-task and multitask settings.

The paper shows that, despite being trained only for next-token prediction, the learned internal representations contain substantial information about the underlying 2D city coordinates, as demonstrated by linear probing. The quality and geometry of these representations depend strongly on the task. The authors further show that multitask training yields more stable and geometrically consistent latent representations. Finally, when a new fictional city is introduced during fine-tuning, the model tends to encode it in task-specific subspaces rather than integrating it into the unified representation learned during pretraining.

**Compliance With Llm Reviewing Policy:**

Affirmed.

**Final Justification:**

I appreciate the additional experiments and clarifications provided in the rebuttal, which strengthen the empirical support for the claims and help address some of my concerns.

In particular, the additional analyses (e.g., multiple representation metrics, wider model, and Atlantis variations) make the observed phenomena more robust within the proposed framework. I also find the discussion connecting these results to broader settings (e.g., vision and continual learning) helpful for contextualizing the work.

That said, my overall assessment remains unchanged. While the synthetic setup is well-motivated and enables controlled investigation, the lack of validation beyond this setting still limits how strongly the conclusions can be extrapolated to more realistic domains. As such, I do not see a strong reason to adjust my score.

Overall, I find the paper technically solid and the rebuttal helpful, but my initial evaluation still stands.

**Key Questions For Authors:**

1. Do the authors expect similar phenomena to arise in other modalities, such as vision? In particular, would comparable convergence and divergence effects appear in image-based world models?
1. The fine-tuning analysis is particularly interesting. If the distinction between forward and backward behavior is real and robust, how might this insight be used in continual learning? For example, could it suggest new training objectives, regularizers, or update constraints that encourage new entities or concepts to be integrated into shared representations rather than isolated task-specific subspaces?

**Limitations:**

yes

**Strengths And Weaknesses:**

Strengths
1. The paper uses a carefully controlled synthetic setup, which allows the authors to isolate the phenomena of interest and present convincing empirical evidence.
1. The results are interesting and easy to interpret. Because the setting is designed to resemble aspects of the real world, the findings are more intuitive and engaging than in many purely abstract synthetic studies.
1. The paper is very clearly written and enjoyable to read. The framing, motivation, and experimental narrative are all presented in an accessible way.

Weaknesses
1. It is somewhat disappointing that the paper contains no experiments on real-world data. Although the synthetic setting is well designed and appropriate for analysis, it remains unclear how strongly the findings transfer to more natural settings.
1. The current study demonstrates these effects in a world model built around 2D geometric structure. However, it is less clear to what extent the conclusions will carry over to settings where the underlying world is less geometric, such as language, or where entities are characterized by properties like color, texture, or shape rather than primarily spatial relations. This limits the current scope of the claims.

---

> ### Author Rebuttal · Authors · 2026-03-31
>
> ## **Response to Reviewer hkRS**
>
> Thank you for the thoughtful review! We are glad you found the **synthetic setup convincing**, the paper **"clearly written and enjoyable to read"** with **"convincing empirical evidence"**.
>
> We ran some additional experiments to improve the paper. In specific we did:
>
> 1. 4 additional representation metrics (CKA, RSA, SNN, SVCCA) on all 1,2,3 task pretrained models.
> 2. An effective rank analysis.
> 3. Scattered Atlantis experiment (1pretraining + (21+7) fine tunings)
> 4. 2x width ablation.
>
> Please find the plots supporting the below at [**[Supplementary Page](https://anonymous.4open.science/r/anonymous-icml-rebuttal-crdt-FE8D/README.md)**].
>
> ---
>
> **W1: No real-world data.** We agree this is a limitation, and extending to other modalities is important future work. The controlled setting lets us make causal claims (manipulating task composition while holding world, architecture, and budget fixed), which observational studies cannot. The phenomena we observe do have parallels in real LMs: the reversal curse (Berglund et al., 2024) and knowledge editing challenges (Lampinen et al., 2025) both involve gradient-based adaptation failing to integrate new information, similar to our divergent task finding. **However,** a synthetic settings allows a much more robust verification of the results we managed to verify the divergent task effect across 4 random seeds, a 2x wider model (Supp. Fig. 4), and with different Atlantis geometries where cities are scattered uniformly across the globe instead of clustered (p=0.0001, Supp. Fig. 3a-b), ruling out several potential confounds.
>
> ---
>
> **W2: Only geometric structure.** We chose a geometric world precisely because it allows us to compare the model's representations to a known ground truth via linear probing. And thus the geometry is less of a domain but more of a tool to verify if a model which never sees coordinates can discover the latent structure. Our findings (convergence, divergent tasks, forward/backward modularity) are about how models learn latent structure from tasks, not about geometry specifically. In fact, to the best of our knowledge, while several works verified the existence of language structure in representations (https://arxiv.org/abs/2210.13382, https://arxiv.org/abs/2310.06824, https://arxiv.org/abs/2410.11767) the study of geometrical representations in sequence models seems relatively limited with the exception of Gurnee &Tegmark 2023.
>
> ---
>
> **Q1: Other modalities?** We could expect similar behavior whenever a model operates on a set of defined entities governed by rules. The broader point is that neural networks can fail to achieve "backward modularity", where they seem to use a shared world model during the forward pass, but gradients from certain tasks fail to update representations consistently with that world model. In vision, tasks like depth estimation and surface normal prediction operate on the same 3D scene but impose different geometric demands, and multi-task vision models show task interference (Standley et al., 2020), and thus we could expect such a phenomena in vision as well.
>
> ---
>
> **Q2: Continual learning?** This is directly our next step. We now have a setup where the model can learn new cities in-context (not just via fine-tuning), and we are able to reproduce the in-context learning vs fine-tuning gap highlighted in recent work (Berglund et al., 2024; Lampinen et al., 2025; Park et al., 2025). The forward/backward modularity distinction suggests concrete directions: (1) search for architectures where ICL (forward representations) and FT (backward in-weights) representation are indistinguishable and (2) screening tasks for divergence before fine-tuning. We believe establishing the phenomenon clearly is a necessary first step, and our framework now enables systematic study of these interventions.
>
> ---
> ---
>
> **Summary:** We thank the reviewer again for the detailed and encouraging feedback! We hope our additional experiments and analyses address the concerns raised, and we would be grateful if you would consider recommending our paper more strongly.
>
> We are happy to answer any follow up questions!
>
> Thank you!

---

> > ### Author Rebuttal · Reviewer_hkRS · 2026-04-04
> >
> > Thank you for the detailed and thoughtful rebuttal. I appreciate the additional experiments and clarifications provided, which strengthen the empirical support for the claims and help address some of my concerns.
> >
> > In particular, the additional analyses (e.g., multiple representation metrics, wider model, and Atlantis variations) make the observed phenomena more robust within the proposed framework. I also find the discussion connecting these results to broader settings (e.g., vision and continual learning) helpful for contextualizing the work.
> >
> > That said, my overall assessment remains unchanged. While the synthetic setup is well-motivated and enables controlled investigation, the lack of validation beyond this setting still limits how strongly the conclusions can be extrapolated to more realistic domains. As such, I do not see a strong reason to adjust my score.
> >
> > Overall, I find the paper technically solid and the rebuttal helpful, but my initial evaluation still stands.

---

> > > ### Author Response · Authors · 2026-04-04
> > >
> > > Thank you for taking the time to read our rebuttal and for the thoughtful acknowledgement.
> > >
> > > We are glad that the additional experiments and analyses helped make the observed phenomena more robust.
> > >
> > > Thank you again for your time and effort reviewing our paper!

---

### Decision · Program_Chairs · 2026-04-30

**Decision:**

Accept (regular)

**Comment:**

This is a carefully designed and well-executed empirical study that advances understanding of how neural representations develop under multi-task learning. While limited to synthetic data, the work is rigorous within its scope and makes findings that bear on important and timely hypotheses, namely the Multitask Scaling Hypothesis and the Platonic Representation Hypothesis, as well as well-known phenomena in real models (e.g., reversal curse). The main strength is the controlled experimental design. As the authors argue in their rebuttal, the setup enables "causal claims (manipulating task composition while holding world, architecture, and budget fixed) which observational studies cannot" make. This argument was well-received by reviewers, who praised the "carefully controlled synthetic setup." The key finding that multi-task training drives representation convergence but certain "divergent" tasks harm fine-tuning adaptation is both interesting, new, and well-established through extensive ablations and across multiple metrics. The authors should work to incorporate much of the new analysis that occurred during the discussion period, which greatly strengthened the paper and ultimately convinced reviewers. The primary limitation of the paper remains its focus on an entirely synthetic setup. However, the authors argued persuasively that this motivates future work, which reviewers agreed with, saying that this "lays a solid foundation for other works to expand on".